



# Moisture and temperature effects on the radiocarbon signature of respired carbon dioxide to assess stability of soil carbon in the Tibetan Plateau

Andrés Tangarife-Escobar[1], Georg Guggenberger[2], Xiaojuan Feng[3], Guohua Dai[3], Carolina Urbina-Malo[2], Mina Azizi-Rad[1], and Carlos A. Sierra[1,4]

[1]Max Planck Institute for Biogeochemistry, Jena, Germany
[2]Institute of Soil Science, Leibniz Universität Hannover, Hannover, Germany
[3]Institute of Botany, Chinese Academy of Sciences, Beijing, China
[4]Department of Ecology, Swedish University of Agricultural Sciences, Uppsala, Sweden

**Correspondence:** Andrés Tangarife-Escobar (atanga@bgc-jena.mpg.de)

**Abstract.** Microbial release of $CO_2$ from soils to the atmosphere reflects how environmental conditions affect the stability of soil organic matter (SOM), especially in massive organic-rich ecosystems like the peatlands and grasslands of the Qinghai-Tibetan Plateau (QTP). Radiocarbon ($^{14}C$) is an important tracer of the global carbon cycle and can be used to understand SOM dynamics through the estimation of time lags between C fixation and respiration, often assessed with metrics such as

age and transit time. In this study, we incubated peatland and grassland soils at four temperature (5, 10, 15 and 20 °C) and two water-filled pore space (WFPS) levels (60 and 95 %), and measured the $^{14}C$ signature of bulk soil and respired $CO_2$. We compare the relation between the $\Delta^{14}C$ of the bulk soil and the $\Delta^{14}CO_2$ of respired carbon as a function of temperature and WFPS for the two soils. To better interpret our results, we used a mathematical model to analyse how the calculated number of pools, decomposition rates of carbon ($k$), transfer ($\alpha$) and partitioning ($\gamma$) coefficients affect the $\Delta^{14}C$-bulk and $\Delta^{14}CO_2$

relation, with their respective mean age and mean transit time. From our incubations, we found that $^{14}C$ from peatland was significantly more depleted (old) than from grassland soil. Our results showed that changes in temperature did not affect the $\Delta^{14}C$ values of respired $CO_2$ in either soil. However, changes in WFPS had a small effect on the $^{14}CO_2$ in grassland soils and a strong influence in peatland soils, where higher WFPS levels led to more depleted $\Delta^{14}CO_2$. In our models, we observed large differences between slow and fast cycling systems, where low values of $k$ modified $\Delta^{14}C$ patterns due to the incorporation of

$^{14}C$-bomb in the soil. Hence, the correspondence between $\Delta^{14}C$ and age and transit time strongly depended on the internal dynamics of the soil ($k$, $\alpha$, $\gamma$ and number of pools) as well as on model structure. We conclude that the stability of carbon in these systems depends strongly on the direction of change in temperature and moisture and how it affects the rates of SOM decomposition. Finally, $\Delta^{14}C$ modelling along with empirical data from SOM dynamics is a useful tool to improve predictions on interactions between terrestrial and atmospheric carbon.





## 1 Introduction

Studying soil organic matter (SOM) stability and persistence in a globally-changing environment is of fundamental importance to understand temporal variations of carbon cycling in the earth-climate system. Soil constitutes the largest carbon (C) stock in the terrestrial biosphere (Chen et al., 2021), making it a key component in global climate models (McGuire et al., 2001; Wieder et al., 2013; Xu et al., 2016). Physical, chemical and biological properties determine SOM decomposition and in consequence

its persistence over decades to millennia (Schmidt et al., 2011). Temperature and moisture are two of the most important abiotic variables controlling the rates of SOM decomposition (Sierra et al., 2015) and its transit time across the different ecosystem pools. Changes in these environmental conditions are, however, occurring simultaneously, highlighting the necessity to conduct multifactorial experiments to disentangle the dominant mechanisms on such cycling times. Hence, discerning the relationship between SOM persistence and C cycling times in terrestrial ecosystems is imperative to improve climate change models and

to inform decisions on $CO_2$ mitigation and management strategies (Bradford et al., 2016; Mesfin et al., 2021).

The Qinghai-Tibetan Plateau (QTP) with an area of 2.5 million $km^2$ is a macroregion of global importance for the cycling of water, carbon and other biogeochemical elements (Anslan et al., 2020). Land cover is mainly dominated by alpine grasslands (44 %) (Scurlock and Hall, 1998; Gao et al., 2014) which store 23.4 % of China's total organic carbon and 2.5 % of the global

soil carbon (Genxu et al., 2002). The QTP also hosts one of the largest high-mountain marshes in the world (Xiang et al., 2009; Chen et al., 2014; Ma et al., 2016), the Zoige peatlands, considered to be the most important carbon stock in peatlands for China (Liu et al., 2018). However, climate change and land cover change are currently driving the net carbon balance through $CO_2$ and $CH_4$ effluxes from the grasslands (Piao et al., 2012; Chen et al., 2014, 2017; Du and Gao, 2020) and the peatlands at both daily and interannual timescales (Hao et al., 2011; Kang et al., 2018; Liu et al., 2019b; Chen et al., 2021). Anthropogenic

activities in the Zoige peatlands (drainage, peat mining and overgrazing) have caused a degradation of approximately 30 % of wet and dry meadows (Chen et al., 2014; Zhou et al., 2021). Additionally, the QTP has been facing an air temperature increase of 0.2°C per decade over the past 50 years (Zhang et al., 2013; Chen et al., 2014; Yang et al., 2014; Ganjurjav et al., 2016), equivalent to two to three times faster than the world average (Yao et al., 2019; Nieberding et al., 2020), along with a moderate increase in precipitation (Dong et al., 2018). According to the IPCC (Arias et al., 2021), climate models predict an increment

in heavy precipitation events and high temperature extremes for this region.

Such changes in temperature and soil moisture control the magnitude in which stabilization and de-stabilization mechanisms enable carbon storage in the QTP (Xiang et al., 2009; Ma et al., 2016). It has been observed that warming increases soil respiration (Rustad et al., 2001; Lu et al., 2013; Pold et al., 2015) and that soil moisture modulates ecosystem and soil respi-

ration (Geng et al., 2012; Piao et al., 2012; Pan et al., 2022; Azizi-Rad et al., 2022). Although the influence of destabilization mechanisms on SOM decomposition has been already studied in the Zoige peatlands (Zhao et al., 2011; Wang et al., 2015; Liu et al., 2016, 2018; Li et al., 2018; Liu et al., 2019a), temporal scales of SOM persistence under temperature and soil moisture variations are not clearly understood yet. Therefore, a quantification of the SOM persistence is of vital importance to predict




climate change feedback magnitudes and pathways.


A useful approach to quantify SOM persistence is through the theory of compartmental dynamical systems, where soil is understood as a set of interconnected pools with transformations of C to different forms (gas, dissolved or solid) (Emanuel et al., 1984; Schimel, 1995; Sierra et al., 2018a). Therefore, C can be characterized by the time it remains inside a compartment or the entire system (Eriksson, 1971; Bolin and Rodhe, 1973) through the calculation of age and transit time. Here, age is defined as

the time elapsed since the C entered the system until the time of observation in the bulk soil, while transit time is defined as the time spent by the C between the entry to the system and its exit as respired $CO_2$ (Eriksson, 1971; Bolin and Rodhe, 1973; Manzoni et al., 2009; Sierra et al., 2018b). These times *(T)* are however, not the same for all the C atoms in a soil, since physical, chemical and biological stabilization and destabilization processes of SOM modify the rate at which mixing and storage occurs.

Radiocarbon measurements in bulk soil and in respired $CO_2$ are a powerful tool to approximate ages and transit times of C in soils as they trace back the trajectory of C through the different stocks on decadal to millennia timescales (Trumbore, 2000; Sierra et al., 2014; Schuur et al., 2016; Estop-Aragones et al., 2020; Chen et al., 2021). By comparing the $^{14}C$ relative abundance in the carbon pools with the atmospheric $^{14}CO_2$ concentrations produced after the nuclear weapon tests during the early 1960's, it is possible to model soil-atmosphere C cycling times (Trumbore, 2000).


$^{14}C$ values can be altered by SOM (de)-stabilization processes and soil characteristics such as soil organic carbon (SOC) content, age and diagenetic state, thaw depth, redox state, seasonality, etc. (Gaudinski et al., 2000; Trumbore, 2000; von Lützow et al., 2008; Sierra et al., 2018b; Estop-Aragones et al., 2020). In recent decades, it has been widely observed that temperature plays a major role on the dynamics of SOM (Knorr et al., 2005; Davidson and Janssens, 2006; Feng and Simpson, 2008) and

the mean age of respired $CO_2$ (Hopkins et al., 2012; Chen et al., 2021) by increasing the decomposition rates from fast-cycling pools (Trumbore et al., 1996) and mobilizing old C in warming conditions (Dutta et al., 2006; Briones et al., 2010). In contrast, other studies have suggested that warming does not lead to release of old C (Briones et al., 2021; Dioumaeva et al., 2002) and that non-labile C decomposition is insensitive to temperature increase (Liski et al., 1999). By comparison, drying phenomena increased the release of modern $^{14}CO_2$ from shallow soil layers but preserved the old soil carbon pools in deeper layers (Kwon

et al., 2019). Observations in rich-SOC soils of the Arctic indicated the release of old C from deep soil layers after thaw and drainage processes (Schuur et al., 2009; Estop-Aragones et al., 2020). Such a process might be occurring in the Zoige peatland soils due to the seasonal presence of frozen layers (Liu et al., 2021; Yang et al., 2022). Nonetheless, the influence of temperature and soil moisture on SOM persistence is a matter of current debate.

Changes in temperature and moisture contribute to the destabilization of carbon in soils from the QTP. Hence, we hypothesize that higher temperature and moisture levels would contribute to increasing the age of respired $CO_2$ for soils subjected to controlled manipulations. Greater ages of respired $CO_2$ would imply that previously stabilized C is destabilized by changes in the manipulated environmental factors. For this purpose, we posed three specific research questions: (1) Do specific changes





in temperature and moisture result in the release of old carbon in the respiration flux? (2) Are there differences in the age of respired $CO_2$ between a grassland and a peatland soil from the QTP? (3) How can radiocarbon data in bulk soil and respired $CO_2$ be interpreted to understand the effect of changes in decomposition rates on ages and transit times? To address these questions, we conducted a controlled multifactorial experiment with soils from two different ecosystems in the QTP, and measured their radiocarbon signature in the bulk soil and the respired $CO_2$. In addition, we used a mathematical model to better interpret our results with the concepts of age and transit time of carbon.

## 2 Materials and methods

To evaluate the influence of soil moisture and temperature on the $^{14}C$ values from bulk soil and respired $CO_2$ of grasslands and peatlands soils, we incubated soil samples from a high elevation grassland at the Nam Co catchment collected in September 2018 and a peatland from the Zoige region sampled in July 2021, both located in the QTP.

### 2.1 Site description and sampling

#### 2.1.1 Nam Co grassland

The Nam Co site (Fig. 1) is located in the central Tibetan Plateau (altitude of the Nam Co lake 4,726 m a.s.l.) and represents a frequent study location for monitoring and tracking of environmental changes over different time scales (Anslan et al., 2020). The dominant climate at Nam Co consists of cold winters and short and moist summers. Mean annual temperature (MAT) measured at the NAMORS research station was -0.6 °C (2006 – 2017) and mean annual precipitation (MAP) was 406 mm, occurring mostly during the monsoon season between May and October (Anslan et al., 2020). The sampling area was mainly covered by *Kobresia pygmaea* and has been grazed intensively by yaks and sheep. A total amount of 40 kg of soil was collected from randomly selected spots at depths between 5 and 15 cm within an area of about 40 m$^2$. The soil was then mixed homogeneously to produce one single sample which consisted of a sandy loam with a bulk density of 1.3 g/cm$^3$, pH of 7.5, 3.2 % of total organic carbon, 0.05 % of inorganic carbon and 0.3 % of total nitrogen, cation exchange capacity of 89 mmol$_c$/kg soil and electrical conductivity of 89.6 $\mu$S/cm.



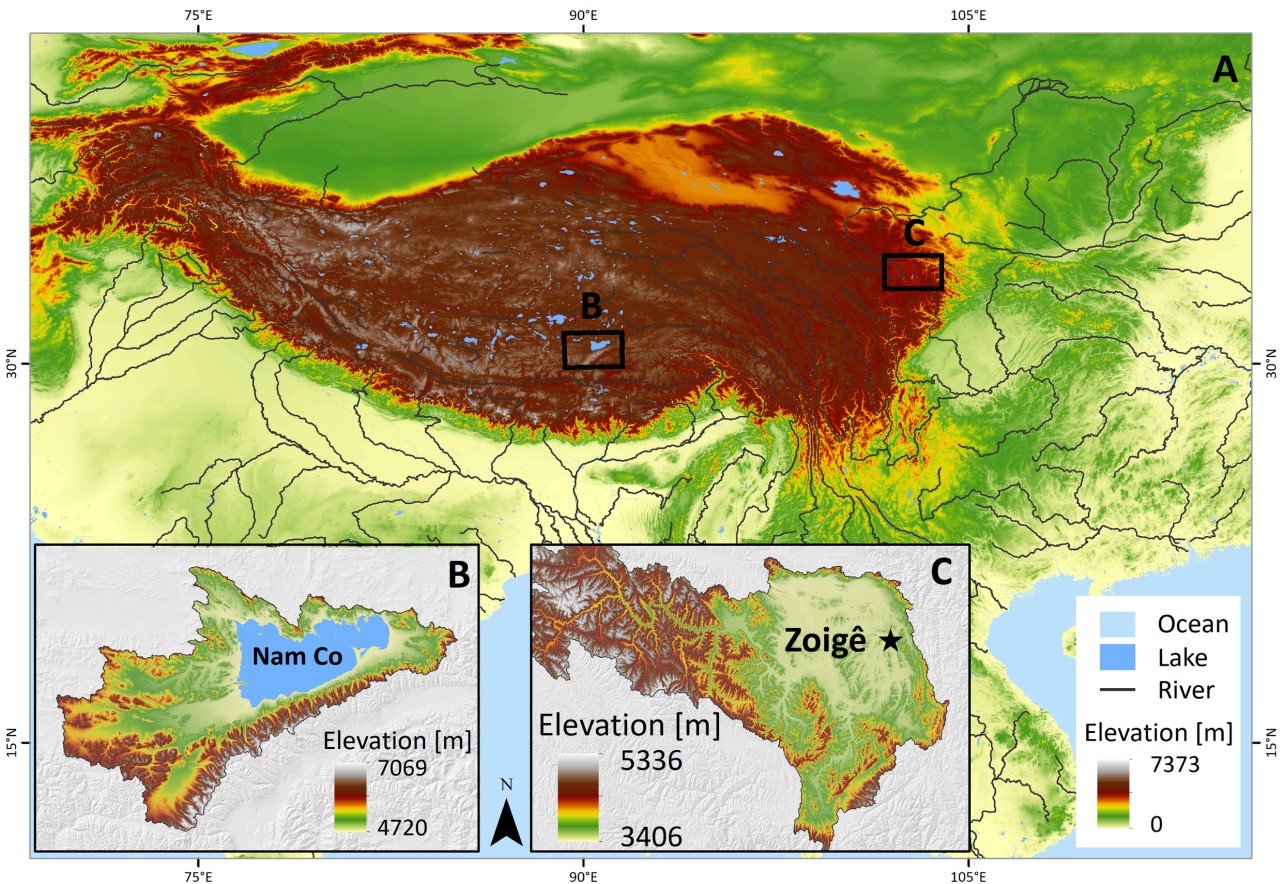

**Figure 1.** a) Overview of the QTP; b) Location of the Nam Co lake with land cover dominated by grasslands, and c) Zoige peatlands highlighting the location of Zoige city. The sampling site is located in the southeastern of the peatland (Ruokeba, Hongyuan county). SRTM elevation data from Jarvis et al. (2008).

### 2.1.2 Zoige peatlands

The Zoige peatlands are located in the northeast of the QTP (average elevation of 3,400 m a.s.l.) and cover an area of about 4,605 km$^2$ on the headwaters of the Yellow River basin. These peatlands have been recognised as one of the largest high-
115 mountain marshes in the world (Xiang et al., 2009; Ma et al., 2016) and store 88 % of the carbon in the QTP (Chen et al., 2014). The Zoige region shows a MAT of 1.5 °C, where the warmest monthly temperature (11 °C) is recorded in July and the coldest in January (-10.1 °C). The MAP is 720 mm, and according to data from the Chinese National Meteorological Information Centre (www.nmic.gov.cn), the region has faced a slight drying trend and a warming of 0.4 °C per decade over the last 40 years (Chen et al., 2014; Yang et al., 2014). The soils are unfrozen from April to October while the layer between 0
120 and 50 cm depth is seasonally frozen between November and March. Vegetation cover consists mainly of *Potentilla anserine, Blysmus sinocompressus, Kobresia myosuroides* and *Scirpus triqueter*. The samples (0 – 35 cm depth) were collected from an



area of 50 m$^2$ at a long-term monitoring site ($33°4'5''N, 102°33'52''E$), then thoroughly mixed to produce a single sample. The soil bulk density was 0.3 g/cm$^3$, contained in average 27.6 % of organic carbon, 0.06 % of inorganic carbon and 1.8 % of total nitrogen.

## 2.2 Incubation of peatland and grassland soils

We conducted two sets of incubation experiments, one set with the grassland soil and a second set with the peatland soil. Samples were incubated with their original roots to minimise disturbance and allow comparisons with field conditions, however stones were removed. Each of the sets were placed at two different WFPS values (60 and 95 %) combined with four temperature levels (5, 10, 15 and 20 °C) for a total of 69 samples (2 sites × 4 temperature levels × 2 moisture levels × 4-5 replicates, to prevent scarcity of data due to eventual failure in $CO_2$ extraction), totaling 33 for grassland and 36 for peatland. $CO_2$ concentrations were measured at intervals of 1 to 2 weeks using a $CO_2$ analyzer LI-COR 6262. Incubations for each subset ended simultaneously until every sample had an estimated concentration of $CO_2$-C in the headspace equivalent to $\geq 0.2$ mg of C, enough for radiocarbon analysis. Grassland samples were incubated between 15 and 67 days, while peatland samples were incubated for 13 days. For sampling headspace air, 50-ml vials were filled with 12 g of soil ($\pm$ 1.5 g) and placed inside 0.5 L glass flasks along with 0.2 ml of water at the bottom of the flask (away from contact with the sample) to avoid possible drying (Dioumaeva et al., 2002); thereafter the flasks were sealed with rubber plugs and screwed with plastic caps. Flasks with samples were flushed with synthetic air ($CO_2$ free) to remove atmospheric $CO_2$. This flushing marked the starting day of the incubations.

## 2.3 Radiocarbon analysis of incubated soils

Respired $CO_2$ accumulated in the headspace of incubation flasks was extracted and purified on a vacuum line, graphitized by Fe reduction in $H_2$ and measured for $\Delta^{14}C$ by Accelerator Mass Spectrometry (Micadas, Ionplus, Switzerland) in the radiocarbon laboratory of the Max Planck Institute for Biogeochemistry in Jena, Germany (Steinhof et al., 2017).

The radiocarbon content reflects the time a C atom has been in the soil since it was fixed by photosynthesis from the atmosphere (Trumbore, 2000). Carbon fixed since the early 1960's has higher $^{14}C$ than carbon fixed previously due to the enrichment by thermonuclear weapons testing. Samples with significant $^{14}C$ positive values show incorporation by the bomb $^{14}C$, presenting $^{14}C/^{12}C$ ratios greater than that of the 1890 wood used as $^{14}C$ standard for pre-bomb periods. Radiocarbon data are expressed as $\Delta^{14}C$ (the deviation in ‰ from 0.95 times the oxalic acid standard in 1950). The values were corrected to a $\delta^{13}C$ value of -25 ‰ for differences in biological mass-dependent fractionation (Stuiver and Polach, 1977). These data are presented as percentage of Modern Carbon (pMC) that can be converted to $F^{14}C$ by dividing it by 100 and later to $\Delta^{14}C$ using equation (1) (Stuiver and Polach, 1977).





$$\Delta^{14}C = \left[F^{14}Ce^{\lambda_C(1950-t)} - 1\right] \times 1000 \, [‰], \tag{1}$$

where $F^{14}C$ is the Fraction Modern, i.e. the ratio of the measured sample normalized to a $\delta^{13}C$ value of -25 ‰, divided by
0.95 times the measured ratio of the Oxalic Acid I standard (OX-I) (Schuur et al., 2016), $\lambda_C$ is the updated radiocarbon decay
constant (equals 1/8267 $[y^{-1}]$), and $t$ is the year of sampling.

## 2.4 SOC decomposition models to predict $\Delta^{14}C$ as a proxy of SOM persistence

The representation of the SOC dynamics has been commonly described through models that can be expressed as systems of
linear differential equations (Manzoni et al., 2009; Sierra et al., 2012; Sierra and Mueller, 2015; Sierra et al., 2017) of the form:

$$\frac{d\mathbf{C}(t)}{dt} = \mathbf{I} + \mathbf{A} \cdot \mathbf{C}(t), \tag{2}$$

where the vector $\mathbf{C}(t)$ is the rate of change of carbon over time in $n$ different pools; the time dependent $n$-dimensional vector
$\mathbf{I}$ represents the total inputs of carbon to each pool; and $\mathbf{A}$ represents the $n \times n$ dimensional matrix with the rates of car-
bon processing for each pool in its main diagonal and the proportion of carbon transferred from one pool to another in the
off-diagonals (Sierra et al., 2012, 2014; Metzler and Sierra, 2018; Sierra et al., 2018b). This mass balance equation has a
radiocarbon counterpart:

$$\frac{d^{14}\mathbf{C}(t)}{dt} = \mathbf{I}_{^{14}\mathbf{C}}(t) + \mathbf{A} \cdot {}^{14}\mathbf{C}(t) - \lambda^{14}\mathbf{C}(t) \tag{3}$$

where $\lambda$ is the radiocarbon decay constant (1/8267 $[y^{-1}]$).

We used a SOC decomposition model to calculate the $\Delta^{14}C$ in bulk and $CO_2$ as well as their equivalent mean age and mean
transit time of a theoretical soil for the year 2019 calibrated with the data set for the Northern Hemisphere Zone 3 (Hua et al.,
2021). The simulated $\Delta^{14}C$ results aimed at reproducing the similar $\Delta^{14}C$ ranges obtained from the incubated soils with the
objective to compare under which settings the models could describe best laboratory or field conditions. For this purpose, we
implemented a two-pool model (one slow and one fast pool) considering two different structures: parallel and series (Fig. 2)
and modified the parameters involved in the soil C processing (Manzoni et al., 2009; Falloon and Smith, 2000) (Table 1). In
the parallel structure, C enters the soil and splits among the two pools according to $\gamma$ and decomposition occurs independently
in each pool according to their respective $k$. In the series structure, C enters only to the fast pool and it is either decomposed
and emitted to the atmosphere or transferred to the slow pool according to the transfer coefficient $\alpha$.





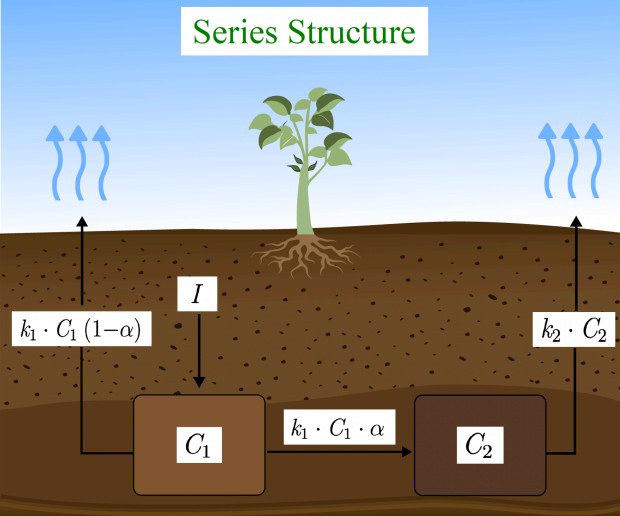

**Figure 2.** Parallel (left) and series (right) model structure and formulas implemented in the simulations. C is fixed into the biomass from atmospheric $CO_2$ through photosynthesis, subsequently it is incorporated into the soil as litterfall. From this state on, it is considered as input ($I$) and will split between the two pools according to the different structures ($\gamma$ for parallel and $\alpha$ for series). The rate at which the C is decomposed ($k$) in each pool ($C$) will depend on microbial activity, environmental factors (temperature and soil moisture) and physical and chemical protection of the SOM (Blanco-Canqui and Lal, 2004; von Lützow et al., 2008; Manzoni et al., 2009). Boxes represent the soil pools, which account with an initial amount of C ($C_1$ =200, $C_2$=5000) and arrows represent directions of inputs and outputs.

Initial parameters of the model such as the initial year of simulation and initial $\Delta^{14}$C values of each pool were considered separately for each type of ecosystem. Moreover, we defined $I$ and $C$ as constant since they can be adjusted depending on the specific soil characteristics. Additionally, we assumed that decomposition rates $k$ reflect the effect of temperature and soil moisture on C cycling timescales (Manzoni et al., 2009), and therefore on $\Delta^{14}$C values. In our approach, only one parameter at time was modified for each simulation.




**Table 1.** Definition of parameters used to evaluate the variation of $\Delta^{14}$C values, mean age and mean transit time and their ranges when type of constant was variable.

| Parameter | Notation | Type | Value | Definition |
|---|---|---|---|---|
| Litter Input | I | Constant | 100 | A scalar or a data.frame object specifying the amount of litter inputs by time |
| Decomposition rate | $k_1, k_2$ | Variable | 0.00001 - 0.8 | A vector of length 2 with the values of the decomposition rate for pools 1 and 2 |
| Partitioning coefficient | $\gamma$ | Variable | 0 -1 | A scalar representing the proportion of I that goes to pool 1 in a parallel structure |
| Transference coefficient | $\alpha$ | Variable | 0 -1 | A scalar with the value of the transfer rate from pool 1 to pool 2 |
| Carbon stocks | $C_1, C_2$ | Constant | 200, 5000 | Initial amount of C for the two pools |

We used the R package SoilR (Sierra et al., 2014) to simulate the temporal dynamics of $\Delta^{14}$C in the bulk soil and the respired $CO_2$ as well as the age and transit time distributions of carbon for two different model structures (parallel and series). Assuming steady-state for the carbon stocks, the probability density function (pdf) of the age (equation 4) and the transit time (equation 5) (Metzler and Sierra, 2018) as well as their means can be calculated by the following expressions (Sierra et al., 2018b):

$$f(a) = -\mathbf{1}^T \cdot \mathbf{A} \cdot e^{a \cdot \mathbf{A}} \cdot \frac{\mathbf{C}^*}{\sum \mathbf{C}^*}, \quad a \geq 0, \tag{4}$$

where $a$ is the random variable age, $\mathbf{1}^T$ is the transpose of the $n$-dimensional vector containing ones, $e^{a \cdot \mathbf{A}}$ is the matrix exponential for each value of $a$, and $\sum \mathbf{C}^*$ is the sum of stocks of all pools at steady state.

The pdf of transit times can be obtained as

$$f(\tau) = -\mathbf{1}^T \cdot \mathbf{A} \cdot e^{\tau \cdot \mathbf{A}} \cdot \frac{\mathbf{I}}{\sum \mathbf{I}}, \quad \tau \geq 0, \tag{5}$$

where $\tau$ represents the random variable transit time.

These pdfs measure the probability that a certain amount of carbon is above or below a specific age or transit time (Sierra et al., 2018b) (equations 6 and 7, respectively). The mean of the age, that is, the expected value of the pdf can be computed by the expression:

$$\mathbb{E}(a) = -\mathbf{1}^T \cdot \mathbf{A}^{-1} \cdot \frac{\mathbf{C}^*}{\sum \mathbf{C}^*}, \tag{6}$$



and the mean of the transit time by

$$\mathbb{E}(\tau) = -\mathbf{1}^T \cdot \mathbf{A}^{-1} \cdot \frac{\mathbf{I}}{\sum \mathbf{I}}. \tag{7}$$

Based on the age and transit time distribution for the different simulated cases, metrics such as the mean, median and
quantiles can indicate the SOM persistence. The relation between ages and transit times may present three cases: *Ta = Tt* (Type
I - well-mixed homogeneous system), *Ta < Tt* (Type II - retention system) and *Ta > Tt* (Type III - non-retention system)
(Sierra et al., 2018b). In the type I, the probability of mineralization and release as $CO_2$ is the same for every C atom. In the
type II, the C is retained for a relatively long time before it is released, as in soils where C is constantly recycled and reused
sequentially before respiration outside the system. In type III, most of the C atoms stay in the system for a short period of
time, but some atoms remain for a long time (Bolin and Rodhe, 1973). Comparisons between these distributions will provide
detailed information on mixing, store, recycling, transport and transformation processes of the SOM (Sierra et al., 2018b).

## 3 Results

### 3.1 $\Delta^{14}$C values of incubated soils under temperature and soil moisture variation

$\Delta^{14}$C values contrasted strongly between grassland and peatland soils (Tangarife-Escobar et al., 2023) (Fig. 3 and 4). Generally,
$\Delta^{14}$C of bulk soil and respiration from the peatland soil was more depleted than from the grassland soil. For example, in
peatland soil the $\Delta^{14}$C values of bulk soil (-85.9 to -60.9 ‰, mean=-80.1, n=36, including outlier of -180) were clearly more
depleted than those of the respired flux (-20.9 to to 23.9 ‰, mean=4, n=36) which indicated that the peatland behaved as a
retention system (type II). In contrast, for the grassland soil, the $\Delta^{14}$C of bulk soil (21.1 to 73.9 ‰, mean=43.3, n=33) was not
too different from the $\Delta^{14}CO_2$ (13.9 to 83.4 ‰, mean=38.5, n=33, including outliers of -227.1 and -105.1) indicating that the
samples behaved mostly as a well-mixed homogeneous system (type I). Such similar values can also reflect a type II system
where $\Delta^{14}$C values in the bulk and the respired flux are equal for the year of sampling (Fig. A7).





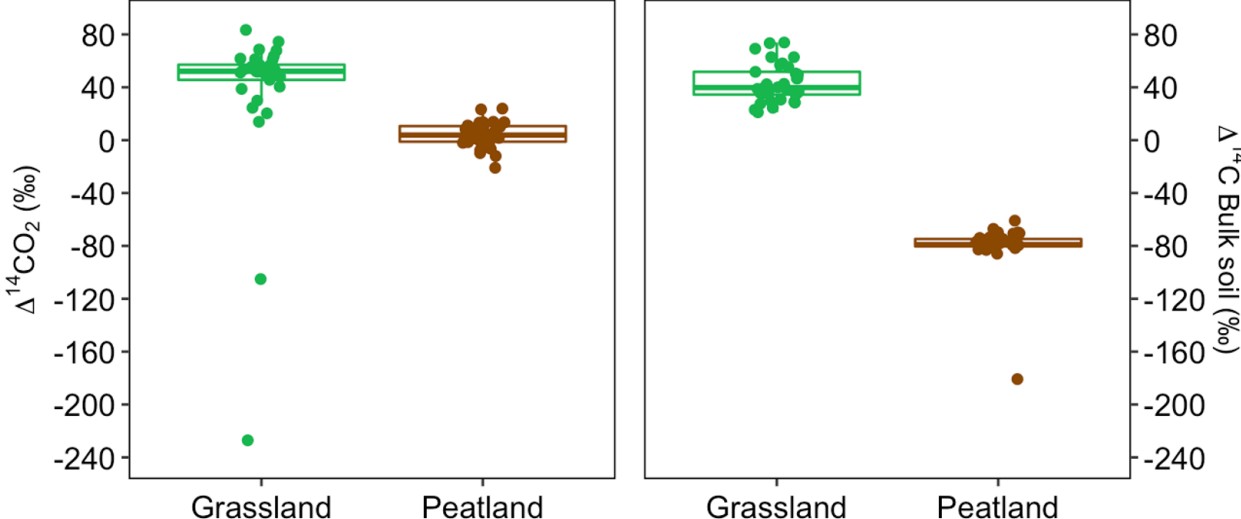

**Figure 3.** Boxplot with the variation of the $\Delta^{14}C$ values in $CO_2$ (left panel) and bulk soil (right panel) for peatland and grassland incubated soils.





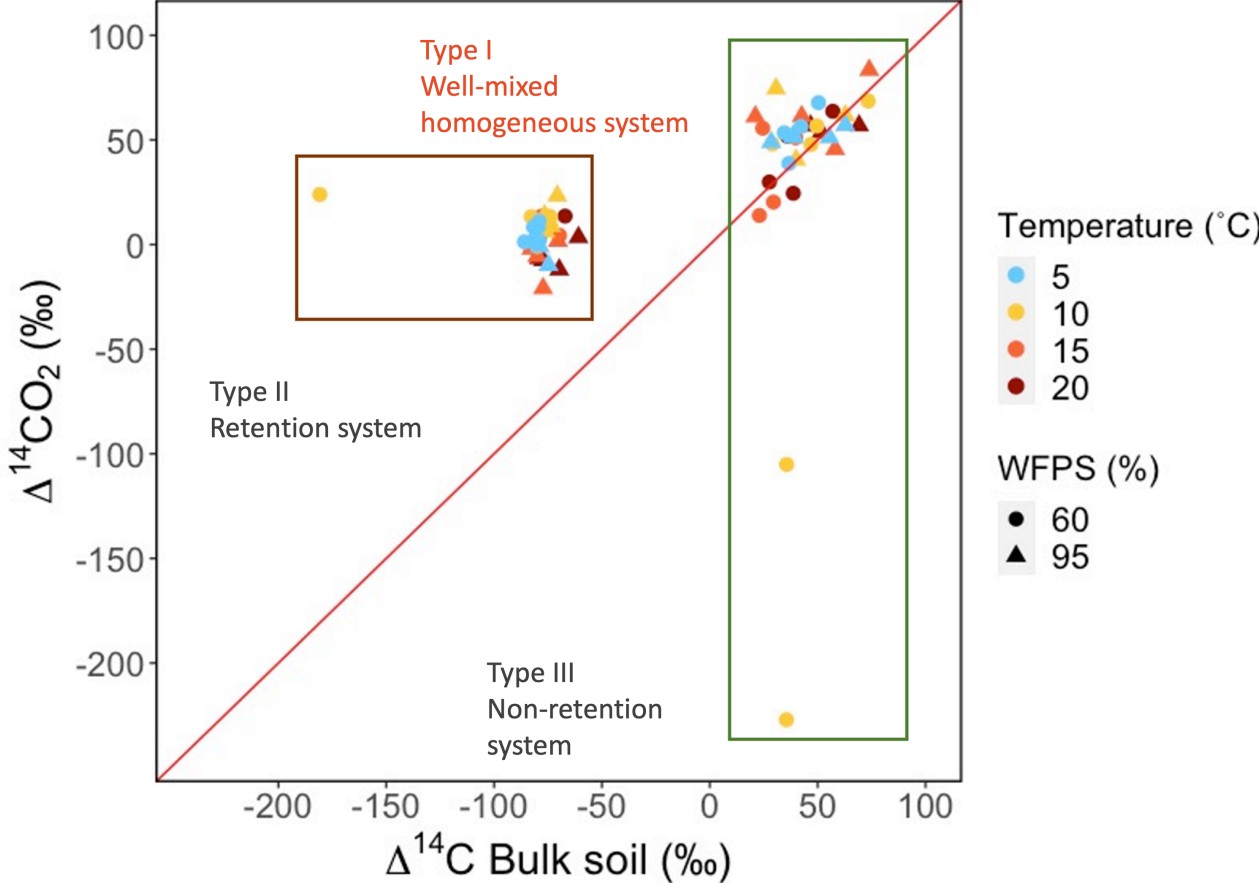

**Figure 4.** Relationship between $\Delta^{14}C$ in bulk soil and $\Delta^{14}C$ in respired $CO_2$ for incubated grassland (green box) and peatland (brown box) soils of the QTP discriminated by temperature and WFPS. Possible types of system according to the $\Delta^{14}C$ relations between bulk and respired carbon.

The temperature treatments did not systematically affect the radiocarbon signature of the bulk or the respired $CO_2$ in peatland or grassland soils (Table 2). There was strong evidence that manipulations in WFPS resulted in changes in the $\Delta^{14}C$ values of bulk soil and $CO_2$ (*p*-values = 0.09 in grasslands and *p*-value = 0.01 in peatlands from an ANOVA test, Table 2, Fig. 5)

except for the bulk soil in peatlands. When the interacting effects of WFPS and temperature were evaluated together, there was no evidence that their interplay affected the radiocarbon signature of both soils (*p*-values > 0.05 for all the analysis). However, higher temperature and WFPS caused an increase of $CO_2$ fluxes from respiration in the treated incubated soils (Table A1). Outliers for bulk and $\Delta^{14}CO_2$ in both peatlands and grasslands occurred in the combined treatment WFPS = 60 and temperature = 10 °C. Such wide variation in $\Delta^{14}C$ both between ecosystems and treatments could be potentially explained by

intrinsic processes affecting soil carbon dynamics, which will be explored in the following sections with a SOC decomposition model.



**Table 2.** Summary of $p$-values obtained from ANOVA tests for the $\Delta^{14}$C of bulk soil and the $\Delta^{14}$CO$_2$ from grassland and peatland soils versus temperature (T) and WFPS independently as well as versus the integrated effect of temperature and WFPS (T · WFPS).

| Ecosystem | Type | T | WFPS | T · WFPS |
|---|---|---|---|---|
| Grassland | $^{14}$C Bulk | 0.8 | 0.091 | 0.21 |
| | $^{14}$CO$_2$ | 0.73 | 0.089 | 0.92 |
| Peatland | $^{14}$C Bulk | 0.4 | 0.21 | 0.65 |
| | $^{14}$CO$_2$ | 0.16 | 0.01 | 0.32 |

**Table 3.** Parameters used for simulations in a SOC decomposition model for fast and slow systems (Tangarife-Escobar et al., 2023).

| System | Fig. | Model structure | $k_1$ | $k_2$ | $\gamma$ | $\alpha$ | Starting year of simulation | $\Delta^{14}$C (‰) for starting year |
|---|---|---|---|---|---|---|---|---|
| Fast cycling | 6 | Parallel | 0.1 - 0.8 | 0.1 | 0.8 | - | 1890 | -4.9 ‰ |
| | | Series | 0.1 - 0.8 | 0.1 | - | 0.8 | 1890 | -4.9 ‰ |
| | 7 | Parallel | 0.8 | 0.001 - 0.8 | 0.8 | - | 1890 | -4.9 ‰ |
| | | Series | 0.8 | 0.001 - 0.8 | - | 0.8 | 1890 | -4.9 ‰ |
| Slow cycling | 8 | Parallel | 0.0001 - 0.8 | 0.0001 | 0.8 | - | 500 | -22.2 ‰ |
| | | Series | 0.0001 - 0.8 | 0.0001 | - | 0.8 | 500 | -22.2 ‰ |
| | 9 | Parallel | 0.1 | 0.000001 - 0.001 | 0.2 | - | 500 | -22.2 ‰ |
| | | Series | 0.1 | 0.000001 - 0.001 | - | 0.8 | 500 | -22.2 ‰ |

## 3.2 Effect of changes of decomposition rates ($k$) on the $\Delta^{14}$C values

Temperature and soil moisture manipulations showed responses in the vertical and horizontal direction in the $\Delta^{14}$C-bulk versus $\Delta^{14}$C-CO$_2$ space. To understand possible drivers of changes in these directions and how they can be interpreted in terms of ages and transit times, we used a SOC decomposition model. We evaluated how model structure, decomposition rates of C, and the partitioning ($\gamma$) and transfer ($\alpha$) coefficients of a two-pool parallel and a series model affected $\Delta^{14}$C values, and as a consequence, the mean age and the mean transit time. Simulations were made for the fast cycling grassland system and the slow cycling peatland (Table 3).

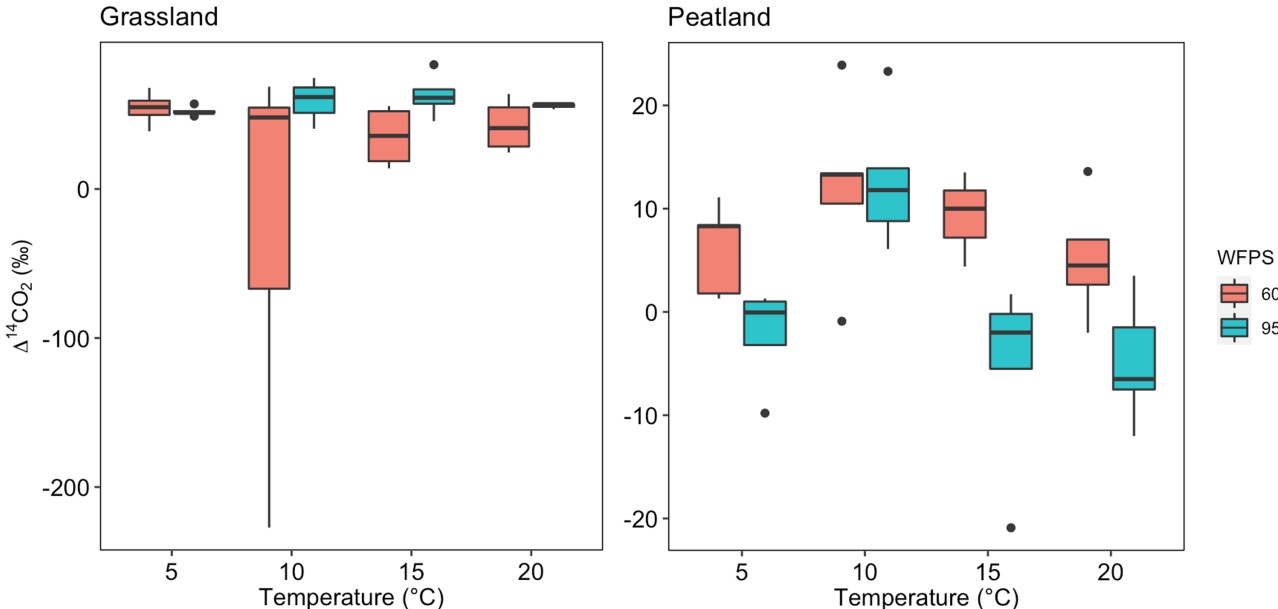

**Figure 5.** Comparison between $\Delta^{14}$C of respiration from incubated peatland and grassland soil at different temperature levels under WFPS = 60 and 95 %. Black points represent minimum and maximum values out of the range between quartile 1 and 3 (25 to 75 % of the data). The quartile 50 (median) represented by the line inside the box indicates the midpoint value in the frequency distribution. Box for the treatment WFPS= 60 % and T= 10 °C shows a large dispersion of the 50 % of the data, which is explained by the outliers observed in Fig. 4.

### 3.2.1 SOC decomposition in fast cycling systems (grasslands)

Variation of $k_1$ yielded $\Delta^{14}$C curves similar to mean age and mean transit time. In other words, in these simulations the change in the parameter $k_1$ resulted in similar trends in $\Delta^{14}$C-bulk versus $\Delta^{14}$C-CO$_2$ space as in the mean age versus mean transit time space ((Fig. 6). In the parallel structure (Fig. 6, A and B), high values of $k_1$ yielded more enriched $\Delta^{14}$C values in the bulk soil than in the respired flux. This was expressed as a fast respiration of SOC and reflected in a short transit time. Simultaneously, low values of $k_2$ resulted in a slow respiration, which was registered in ages longer than transit times, at the initial state of the C

stay in the system. However, as $k_1$ decreased and approached $k_2$, mean age and mean transit time became similar and converged to the 1:1 line. In the case of the series structure (Fig. 6, C and D), the total amount of inputs traveled first through the fast pool, which meant that while $k_1$ is high, most of the C decomposition occurs under the dynamics of the fast pool. Hence, the



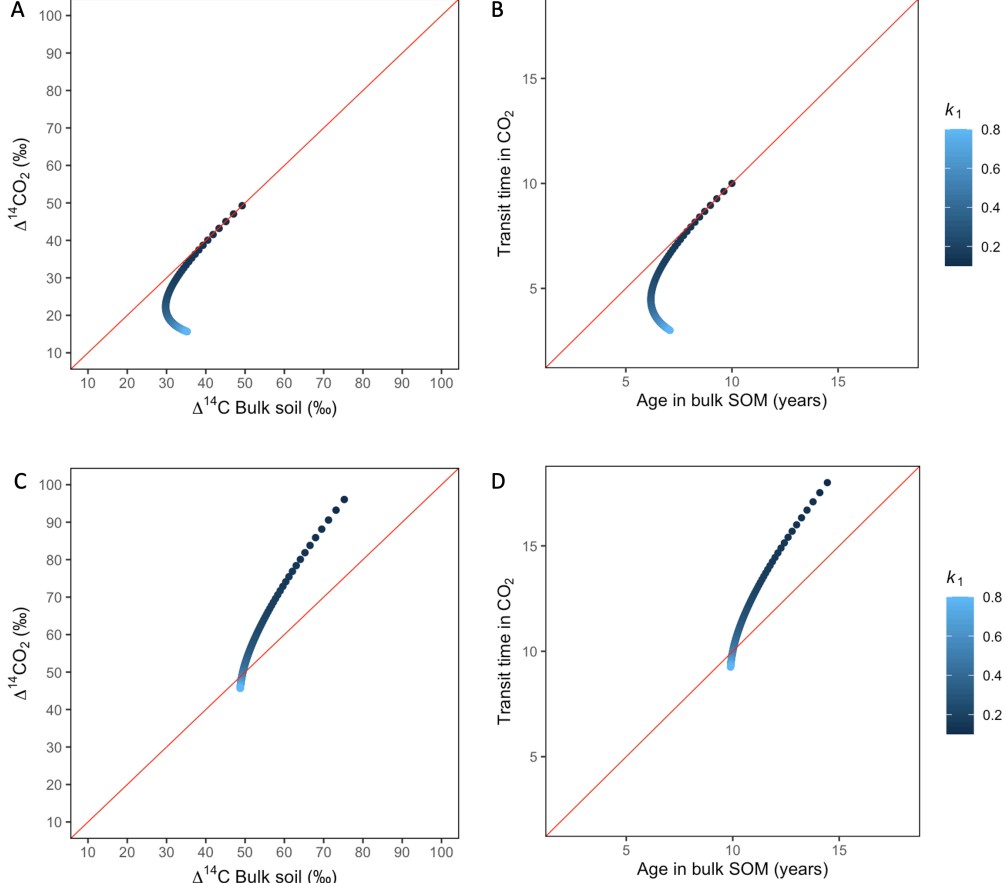

**Figure 6.** Predictions of $\Delta^{14}$C in bulk soil vs $\Delta^{14}$CO$_2$ with their equivalent simulation of mean age in bulk soil vs mean transit time in CO$_2$ for parallel (panel A and B) and series model structure (C and D). Variation of $k_1$ with $\alpha$ = 0.8 and $\gamma$ = 0.8. Complementary prediction with $\alpha$ = 0.1 and $\gamma$ = 0.1 can be seen in Fig. A2.

system showed equal mean age and mean transit time. As $k_1$ decreased, the transfer from the fast to the slow pool becomes more relevant, then the mean transit time increased compared to the mean age.


The behavior of $\Delta^{14}$C values contrasted significantly with that of mean age and mean transit time when $k_2$ tended to lower values (Fig. 7). Generally, when $k_2$ equals $k_1$, $\Delta^{14}$C and mean age and mean transit time showed similar values. However, as $k_2$ decreased, the C stayed in the slow pool for longer time and the $\Delta^{14}$C enriched in the bulk soil until reaching a peak (112 and 89 ‰ for series and parallel structures, respectively), from where it subsequently depleted to the initial $\Delta^{14}$CO$_2$ value. For the parallel structure (Fig. 7, A and B), transit time kept increasing slowly since most of the C stayed in the fast pool for a short time, so, the C remaining in the slow pool contributed to increase the mean age. As for the series structure, most of






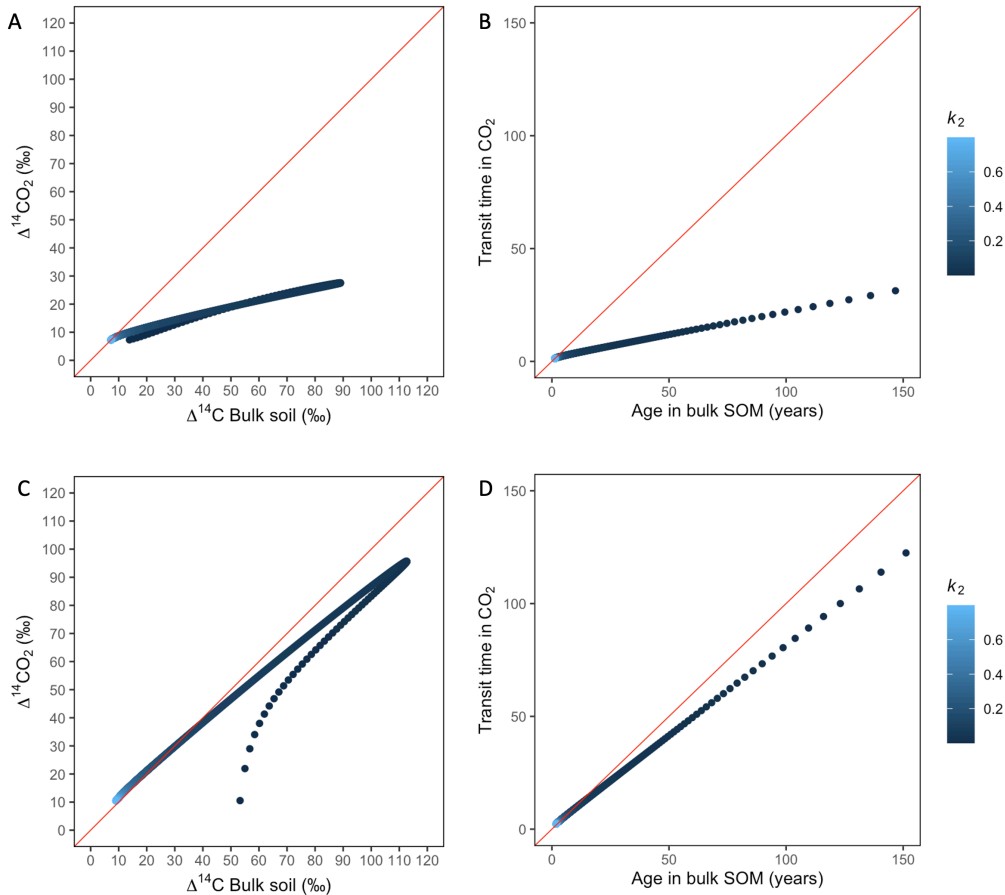

**Figure 7.** Predictions of $\Delta^{14}$C in bulk soil vs $\Delta^{14}$CO$_2$ with their equivalent simulation of mean age in bulk soil vs mean transit time in CO$_2$ for parallel (panel A and B) and series model structure (C and D). Variation of $k$ with $\alpha = 0.8$ and $\gamma = 0.8$.

the C was transferred from the fast to the slow pool due to the high transfer rate $\alpha$, which contributed to increase the transit time.

### 3.2.2 SOC decomposition in slow cycling systems (peatlands)

Simulations of $\Delta^{14}$C values in slow cycling systems resulted in significantly more depleted values in both bulk soil and respired CO$_2$ (Fig. 8) than in fast cycling systems. Generally, $\Delta^{14}$C values were more depleted in the bulk soil than in the respiration flux. Also, the results from these simulations showed very different patterns in the $\Delta^{14}$C-bulk versus $\Delta^{14}$C-CO$_2$ space than in the mean age versus mean transit time space; only with the exception of equal $k_1$ and $k_2$, which resulted in similar $\Delta^{14}$C values as well as similar mean age and mean transit time in the parallel structure.






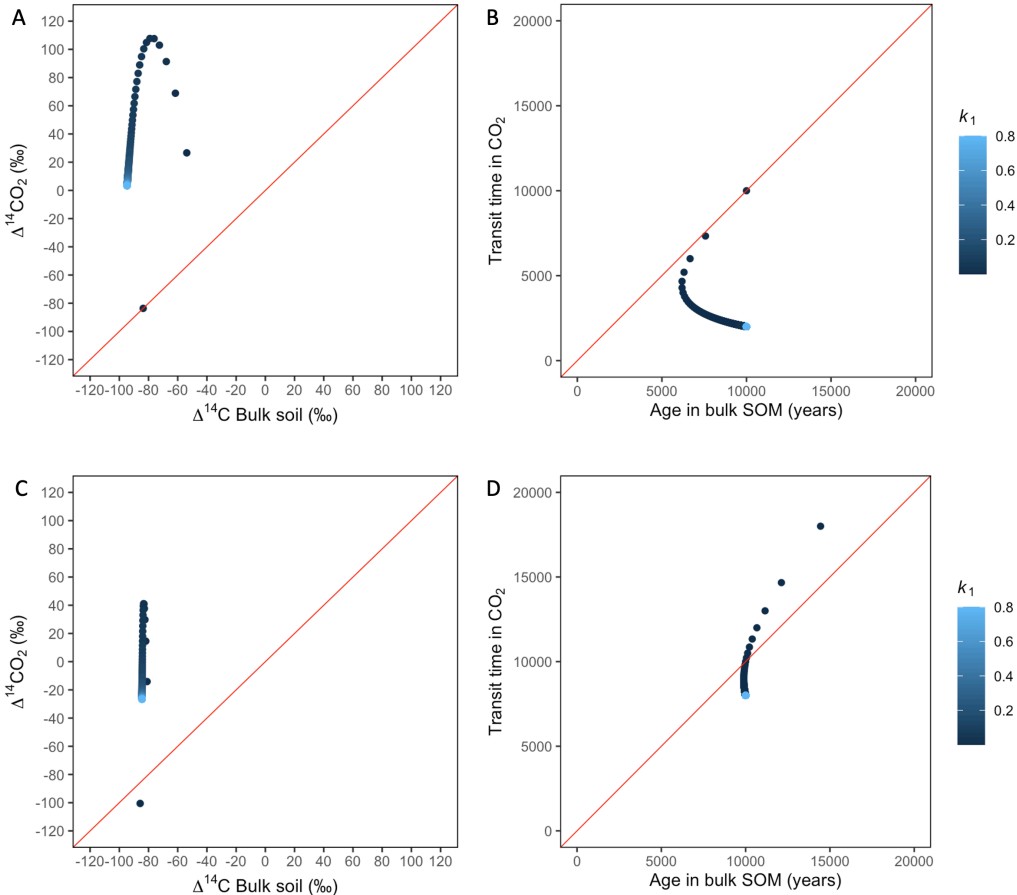

**Figure 8.** Predictions of $\Delta^{14}$C in bulk soil vs $\Delta^{14}$CO$_2$ with their equivalent simulation of mean age in bulk soil vs mean transit time in CO$_2$ for parallel (panel A and B) and series model structure (C and D). Variation of $k_1$ with $\alpha$ = 0.8 and $\gamma$ = 0.8.

Additionally, we looked at the variation of $\Delta^{14}$C keeping the same $k$ values but reducing $\alpha$ and $\gamma$ to 0.1. Simulations indicated that the behavior of $\Delta^{14}$C values were opposite between series and parallel structures when inputs to each pool were inversely proportional (Fig. 7 versus Fig. A3 and Fig. 8 versus Fig. A4). For example, for the case of low values of $k_1$ and $k_2$ (Fig. 9) ($\gamma$ = 0.2 and $\alpha$ = 0.8) both model structures showed similar patterns due to relatively similar amounts of inputs going to the slow pool. When $\gamma$ = 0.8, $\Delta^{14}$C showed values out of the target range (-20 to 23 for $\Delta^{14}$C-CO$_2$ and -86 to -60 for $\Delta^{14}$C-bulk based on incubation results) since decomposition occurred as in a fast cycling system. As $k_2$ decreased, $\Delta^{14}$CO$_2$ enriched and $\Delta^{14}$C and mean age remained higher than mean transit time since the slow decomposition of the slow pool dominated the system response.



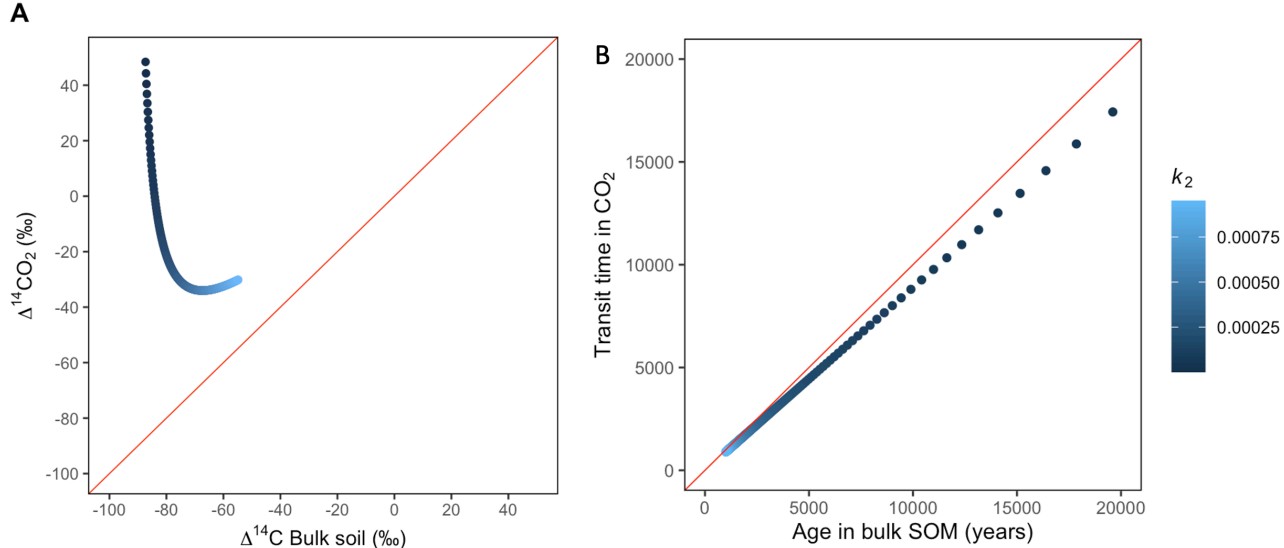

**Figure 9.** Predictions of $\Delta^{14}C$ in bulk soil vs $\Delta^{14}CO_2$ with their equivalent simulation of mean age in bulk soil vs mean transit time in $CO_2$ for parallel (panel A and B). Series model presented exactly the same behaviour. Variation of $k_2$ with $\alpha = 0.8$ and $\gamma = 0.2$.

### 3.3 Variation in the proportion of inputs $\gamma$ and $\alpha$ modulated mean age and mean transit time

Our simulations showed the importance that the partitioning and transference coefficients ($\gamma$ and $\alpha$) have to define the amount of C entering to each pool and in consequence on the mean age and mean transit time (Fig. A1). For a parallel structure, when the total amount of inputs entered only to the fast pool ($\gamma$ =1), the system behaved as a one pool system and mean age and mean transit time were equal. In the extreme opposite ($\gamma = 0$), all C entered only to the slow pool and mean age and mean transit time converged, although with a longer transit through the system. In contrast, for a series structure, the extreme value $\alpha = 1$ resulted in a longer mean transit time than mean age as the contribution to the respired flux comprehended the sum of the decomposition of the newly absorbed C and the initial stocks stored in the fast pool. In the case of slow cycling systems (Fig. A5 and Fig. A6), $\Delta^{14}CO_2$ moved in a wider range including very depleted values (-100 to 70 ‰) and approached to a steady





state of $\Delta^{14}$C in bulk soil from the right when $k_1 = 0.01$ or from the left when $k_1 = 0.1$.


## 4 Discussion

### 4.1 Did changes in temperature and soil moisture result in the release of old carbon in the respiration flux?

We measured the $\Delta^{14}$C values of respired $CO_2$ from soils incubated at different temperatures and WFPS levels to assess whether changes in these variables would result in destabilization of stored C. Our results showed that changes in temperature

did not affect systematically the radiocarbon values of respired $CO_2$ in any of the incubated soils. Nevertheless, changes in WFPS had a moderate effect on the $\Delta^{14}CO_2$ in grasslands, but a strong influence on the $\Delta^{14}CO_2$ of peatlands. Our results indicated that higher WFPS levels led to depleted $\Delta^{14}CO_2$ values in peatlands at all temperature treatments, except at 10 °C. In general, higher temperature and WFPS levels resulted in increased $CO_2$ fluxes although without showing a relationship with the $\Delta^{14}$C values.


In contrast, higher WFPS resulted in more enriched $\Delta^{14}CO_2$ values for grasslands. This suggests that the direction of changing WFPS depends on ecosystem characteristics. It is interesting to observe that the $\Delta^{14}$C in the bulk soil was always more negative than the $\Delta^{14}CO_2$ in the peatlands, while the grasslands registered similar $\Delta^{14}$C values in both states. This indicates that peatlands are systems that stabilize organic matter over time and release it once the stable conditions change. In contrast,

for grasslands two interpretations may arise, first that there was not organic matter stabilization under the incubated conditions since the respired flux reassembles the bulk soil $\Delta^{14}$C signature, or second that the incubation conditions were not strong enough to destabilize the existing old SOM. Our simulations indicate that similar $\Delta^{14}$C values in the bulk and in the respired $CO_2$ can occur at an specific year (Fig A7) independently of the different SOM cycling times in the soil pools.

In tundra ecosystems, Kwon et al. (2019) suggested that drainage of shallow soil layers may have accelerated old carbon decomposition. In addition, Estop-Aragones et al. (2020) concluded that old C would increase in proportion from "cold" across "warm wet" to "warm and dry" for high arctic tundra as well as from "undisturbed" to "burnt active layer" for peatland plateau. These previous findings are in disagreement with our observations since after our experiments, lower WFPS resulted in the release of relatively enriched $\Delta^{14}CO_2$ instead of releasing old carbon. One potential explanation for this discrepancy is that

the existing old SOM could have reacted too slow to drier conditions (WFPS = 65) compared to the fast response of "younger" SOM, causing an evident prevalence of young carbon given the short duration of the incubations. Consistent with this, it has been found that some SOM components or compartments may be more sensitive to modified conditions than others (Feng and Simpson, 2008) and therefore, transit time would react strongly when such sensitivity is higher in the slower pool (Manzoni et al., 2009).




For this reason, our results are indicative, but not conclusive, about the influence of environmental factors on the $\Delta^{14}$C signature of respired carbon. It is possible that our $\Delta^{14}$C values are limited by the short time of the incubations since the dynamics of C transfers cannot be properly observed in short timescales (Crow and Sierra, 2018). For example, during a 1-year incubation experiment, most of the $CO_2$ was derived from labile SOM as the temperature increased (Leifeld and Fuhrer, 2005).

Additionally, other factors such as yearly climate seasonality, daily freeze-thaw cycles and water table oscillation were not replicated in the laboratory due to equipment and time availability. Finally, although temperature has been found to be the main modulator of SOC decomposition rates (Azizi-Rad et al., 2022), dominating over the effect of soil moisture and oxygen availability, it is important to consider that an increase in respiration rates, as observed in our incubations, does not necessarily involve the release of old or young carbon.


## 4.2   Are there differences in the age of respired $CO_2$ between grassland and peatland?

We found that $\Delta^{14}$C values from peatland were strongly more depleted than those in grasslands, indicating the presence and respiration of older carbon in peatlands. Generally, low temperature, low soil microbial activity and anoxic conditions (Xiang et al., 2009; Ma et al., 2016) posed favourable conditions for the stabilization of organic matter since the early Holocene in

the Zoige region (Chen et al., 2014; Sun et al., 2017). Older bulk soil than $CO_2$ demonstrated that the peatland soil behaved as a retention system (Type II) where C was stored for relatively long time before its respiration (Sierra et al., 2018b). The release of that old C may occur when the SOM destabilization dominates over the stabilization mechanisms, for example when peatlands are drained (increasing oxidation through water table reduction) and their temperature is raised (Dutta et al., 2006; Hicks Pries et al., 2013; Lupascu et al., 2014; Estop-Aragonés et al., 2018), both factors mimicked in our incubations.


In contrast, $\Delta^{14}CO_2$ from grasslands were mostly similar to $\Delta^{14}$C in bulk soil, indicating that the soil behaved predominantly as a well-mixed homogeneous system (Type I) where most of the SOM has the same probability of being mineralized and released as $CO_2$ (Sierra et al., 2018b). However, $\Delta^{14}$C relation also indicated that the grasslands shift from retention to non-retention system and vice-versa. In grasslands, SOM from the topsoil is permanently under changing conditions (daily

temperature and soil moisture variation, grazing and mechanical alteration, among others) that promote its fast cycling instead of its stabilization (Han et al., 2017). These factors, along with climatic, vegetation and edaphic properties caused a low SOM stability expressed in a large proportion of labile carbon (Hou et al., 2021). Nonetheless, although SOM accumulation still occurs in grassland areas (Tian et al., 2009) registered in the extremely depleted outliers from the grassland soil respiration, it happens at a slower rate than in peatlands. As a result, most of the fixed atmospheric C is respired at short time scales, which

is in turn registered as young ("post bomb") $\Delta^{14}$C values.

Taken together, SOM stability and ecosystem characteristics defined the relation between $\Delta^{14}$C in bulk soil and $\Delta^{14}CO_2$, which in turn explained the difference in the age of respiration between grassland and peatlands. The $\Delta^{14}CO_2$ values obtained in our incubated peatlands indicate that such C was captured through photosynthesis during a time when the $^{14}$C levels of





the atmosphere were depleted and have remained stored in the soil for a long enough time for radioactive decay to become

relevant. However, it is imprecise to attribute an specific year to the entire bulk soil or $CO_2$ fractions, since they are formed by

SOM accumulated in multiple steps and of different qualities and their respective decomposition.

In that sense, a $\Delta^{14}C$ value is not per se indicative of age or transit time of SOC. To disentangle the utility of radiocarbon

as a tool for SOM persistence in soil and be able to shed light on the time-scales of C cycling in different ecosystems, we used

SOC decomposition models. Our models contribute to understand the time component of SOM persistence through the use of

mean transit time as a parameter that integrates processes of SOM dynamics (Manzoni et al., 2009).

### 4.3 How can $\Delta^{14}C$ values be interpreted to understand the effect of decomposition rates on mean age and mean

transit time?

We simulated the $\Delta^{14}C$ values and the mean age and mean transit time of soils with high and low decomposition rates. Our sim-

ulations were able to reassemble the $\Delta^{14}C$ values obtained from incubations and showed that modelled $\Delta^{14}C$ values differed

widely between slow cycling systems and fast cycling systems. Generally, decomposition rate, transfer rates and partitioning

coefficients of a given model structure modulated $\Delta^{14}C$ values, and in consequence, mean age and mean transit time (Bruun

et al., 2004; Manzoni et al., 2009). For example, fast cycling systems with high decomposition rates resulted in a fast respi-

ration represented by short mean transit times (lower than 20 years). At the same time, low decomposition rates resulted in

longer mean transit times, which in turn represented older respired $CO_2$ (lower than 140 years). In slow cycling systems, where

the decomposition rates were lower, transit times were consequently longer (lower than 20000 years). Independently of the $k$

values, the series model structure increased the transit time of C compared to parallel structure due to the fact that a proportion

of the C atoms had to pass through the two pools before being respired.

Other factors such as observation time and starting year of the simulation modified the response of $\Delta^{14}C$. This is due to the

radioactive nature of the $^{14}C$ isotope and the influence of the "bomb $^{14}C$". For instance, the $\Delta^{14}C$ enrichment during the last

60 years was clearly observed in fast cycling systems only with low $k_2$. Furthermore, the inflexion points on the $\Delta^{14}C$ curves

occurred at different values for slow and fast cycling systems. Hence, we could assume that a simulation with starting year

after 1962 would not show an inflexion point in the $\Delta^{14}C$ due to the absence of the bomb peak.

Our modelling results suggest that the increase of decomposition rates contribute to the release of older carbon in the respired

flux as we hypothesised, but depending on the initial state of the system. This interpretation is not straightforward since the

stability of SOM depends on the specific combination of temperature and moisture for different ecosystems.For peatlands, only

drier conditions and consequent increase of oxygen might increase $k$ and therefore cause the destabilization of SOM. In con-

trast, grasslands would need an increase of temperature to facilitate SOM decomposition provided that moisture and oxygen

are available (Azizi-Rad et al., 2022). SOM decomposition rates are expected to increase at higher temperatures (Leifeld and





Fuhrer, 2005) and towards the extremes of the moisture range (Sierra et al., 2015). However, the destabilization of SOM can

occur in any direction of WFPS variation (depending on the oxygen content) and in turn affect the age and transit time of C in a nonlinear pattern.

From our results we could observe how the relation $\Delta^{14}$C-bulk versus $\Delta^{14}$C-CO$_2$ properly represented the relation mean age versus mean transit time. We found that there is a good correspondence between both relations in the fast cycling systems

as long as the decomposition rates $k$ remain high. Such correspondence did not occur as $k$ became smaller (typical for slow cycling systems) since the appearance of the bomb peak may have introduced anomalies that modified the equivalence between the two relations. In addition, the outlier $\Delta^{14}$C values found in our experimental data, where $\Delta^{14}$C changed drastically in the vertical and the horizontal direction, may be related to this complex response of the $^{14}$C tracer.

To quantify cycling times of carbon, radiocarbon can be used as a tool to understand SOM destabilization and persistence through the use of the concepts of age and transit time and their mutual relation. Nonetheless, it is essential to couple $\Delta^{14}$C measured values with a model that involves the dynamics of soil carbon in different pools and their interaction with the environment. Therefore, the acquisition of empirical data from soils (number of pools, $I$, $C$, $k$s, $\gamma$, $\alpha$) along with the correct setting of model structure will improve predictions on terrestrial and atmospheric carbon interactions.


## 5 Conclusions

Based on the incubation results of soils from the QTP, we showed that the $\Delta^{14}$C values of the peatland are significantly more depleted than the ones of the grassland both in the bulk soil and the respired CO$_2$. Our results indicated that changes in temperature did not affect systematically the radiocarbon values of respired CO$_2$ in any of the incubated soils. Nevertheless, changes

in WFPS had a relatively small effect on the $^{14}$CO$_2$ in grasslands, but a strong influence on the $^{14}$CO$_2$ of peatlands, where higher WFPS levels led to more depleted $^{14}$CO$_2$ values except at 10 °C. In peatlands, more depleted $^{14}$C-bulk values than $^{14}$C-CO$_2$ indicated that SOM stabilizes over time and it is released once the stable conditions change. In grasslands, similar $\Delta^{14}$C values in bulk and respired CO$_2$ indicated that the soil behaved as a well-mixed homogeneous system due to either an absence of SOM stabilization or that the manipulation treatments were not long or strong enough to destabilize the existing

old SOM. In this sense, the short duration of our incubations might have been an obstacle to register the influence of long term factors such as climate seasonality and water table oscillation on SOM dynamics.

From our modelling approach, we conclude that radiocarbon can be used as a tool to understand SOM persistence through the use of the concepts of mean age and mean transit time and their mutual relation. Our simulations were able to reassemble the

$\Delta^{14}$C values obtained from incubations and showed that modelled $\Delta^{14}$C values differed widely between slow cycling systems and fast cycling systems. We found that low values of $k$, more common in slow cycling systems, modified the behaviour of



$\Delta^{14}$C patterns due to the incorporation of $^{14}$C-bomb in the soil system. Hence, the correspondence between these mutual relations strongly depended on the internal dynamics of the soil and its interaction with the environment. For this reason, the acquisition of empirical data from soils (number of pools, $I$, $C$, $k$s, $\gamma$ and $\alpha$) along with the correct setting of model structure will improve predictions on terrestrial and atmospheric carbon interactions.

*Code and data availability.*  Lab analysis results and code are stored at https://doi.org/10.5281/zenodo.7620008 (DOI:10.5281/zenodo.7620008) (Tangarife-Escobar et al., 2023).

*Author contributions.*  The conceptualization, data curation and formal analysis were done by ATE and CS. The investigation was performed by ATE with support from GD and MAR. Methodology and validation were done by ATE and CS. ATE conducted the validation, visualization and writing of the original draft (the latter with the help of CUM). Supervision was done by GG, XF and CS. ATE wrote the manuscript with contributions from CS, GG, XF, CUM, GD and MAR.

*Competing interests.*  The authors declare that they have no conflict of interest.

*Acknowledgements.*  We thank all colleagues who contributed to this study, especially Eike Reinosch for providing the location maps used in Fig.1. and Paula Sierra for the digitization of the scheme used in Fig. 2. Special thanks to David Martini for his helpful advices to improve code writing and to Manuel Röst and Axel Steinhof for their useful training at the $^{14}$C laboratory. Finally, thanks to Nicole Börner for the active collaboration with the project administration and to Susan Trumbore for her feedback on the final version of this manuscript. This study was developed as part of the International Research Training Group (GRK 2309/1) Geo-ecosystems in transition on the Tibetan Plateau (TransTiP) funded by the Deutsche Forschungsgemeinschaft (DFG). The Max Planck Institute for Biogeochemistry provided permanent administrative support.



## Appendix A: Appendix A

**Table A1.** Mean daily $CO_2$ respiration for incubated soils under temperature and WFPS variation.

| Ecosystem | Temperature (°C) | WFPS (%) | Mean $CO_2$ respiration (mg $CO_2$ g soil $^{-1}$ day $^{-1}$) | $\sigma$ |
|---|---|---|---|---|
| Grassland | 20 | 95 | 0.510 | 0.035 |
| | 20 | 60 | 0.371 | 0.016 |
| | 15 | 95 | 0.330 | 0.008 |
| | 15 | 60 | 0.234 | 0.004 |
| | 10 | 95 | 0.209 | 0.011 |
| | 10 | 60 | 0.134 | 0.023 |
| | 5 | 95 | 0.066 | 0.003 |
| | 5 | 60 | 0.044 | 0.001 |
| Peatland | 20 | 95 | 0.040 | 0.001 |
| | 20 | 60 | 0.028 | 0.003 |
| | 15 | 95 | 0.024 | 0.004 |
| | 15 | 60 | 0.016 | 0.003 |
| | 10 | 95 | 0.011 | 0.002 |
| | 10 | 60 | 0.008 | 0.001 |
| | 5 | 95 | 0.005 | 0.000 |
| | 5 | 60 | 0.004 | 0.001 |





**Table A2.** Parameters used for SOC decomposition models in grassland (fast cycling) and peatland (slow cycling).

| System | Fig. | Model structure | $k_1$ | $k_2$ | $\gamma$ | $\alpha$ | Starting year of simulation | $\Delta^{14}C$ (‰) for starting year |
|---|---|---|---|---|---|---|---|---|
| Fast cycling | A1 | Parallel | 0.8 | 0.1 | 0 - 1 | - | 1890 | -4.9 ‰ |
| | | Series | 0.8 | 0.1 | - | 0 - 1 | 1890 | -4.9 ‰ |
| | A2 | Parallel | 0.1 - 0.8 | 0.1 | 0.1 | - | 1890 | -4.9 ‰ |
| | | Series | 0.1 - 0.8 | 0.1 | - | 0.1 | 1890 | -4.9 ‰ |
| | A3 | Parallel | 0.8 | 0.0001 - 0.8 | 0.1 | - | 1890 | -4.9 ‰ |
| | | Series | 0.8 | 0.0001 - 0.8 | - | 0.1 | 1890 | -4.9 ‰ |
| Slow cycling | A4 | Parallel | 0.0001 - 0.8 | 0.0001 | 0.2 | - | 500 | -22.2 ‰ |
| | | Series | 0.0001 - 0.8 | 0.0001 | - | 0.1 | 500 | -22.2 ‰ |
| | A5 | Parallel | 0.01 | 0.0001 | 0 - 1 | - | 500 | -22.2 ‰ |
| | | Series | 0.01 | 0.0001 | - | 0 - 1 | 500 | -22.2 ‰ |
| | A6 | Parallel | 0.1 | 0.0001 | 0 - 1 | - | 500 | -22.2 ‰ |
| | | Series | 0.1 | 0.0001 | - | 0 - 1 | 500 | -22.2 ‰ |




**- $\Delta^{14}C$ variation in fast cycling systems**

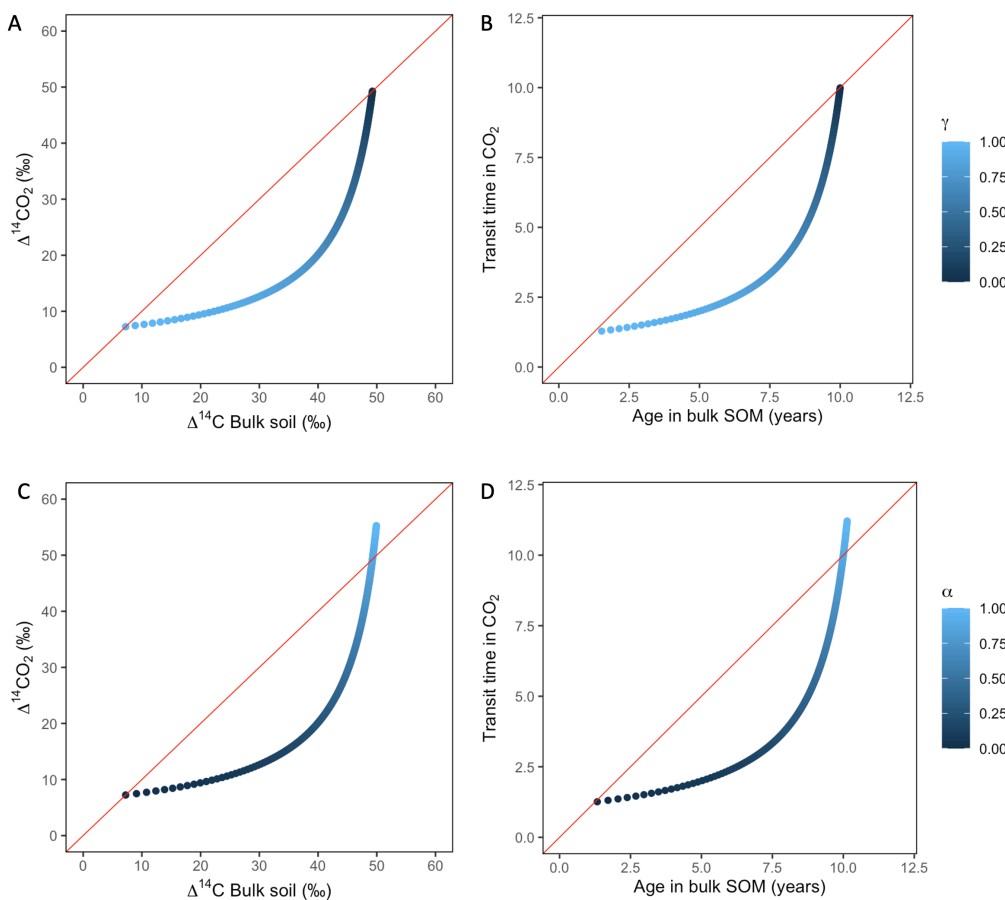

**Figure A1.** Predictions of $\Delta^{14}C$ in bulk soil vs $\Delta^{14}CO_2$ with their equivalent simulation of mean age in bulk soil vs mean transit time in $CO_2$ for parallel (panel A and B) and series model structure (C and D). $k_1 = 0.8$, $k_2 = 0.1$, with $\alpha = 0 - 1$ and $\gamma = 0 - 1$.





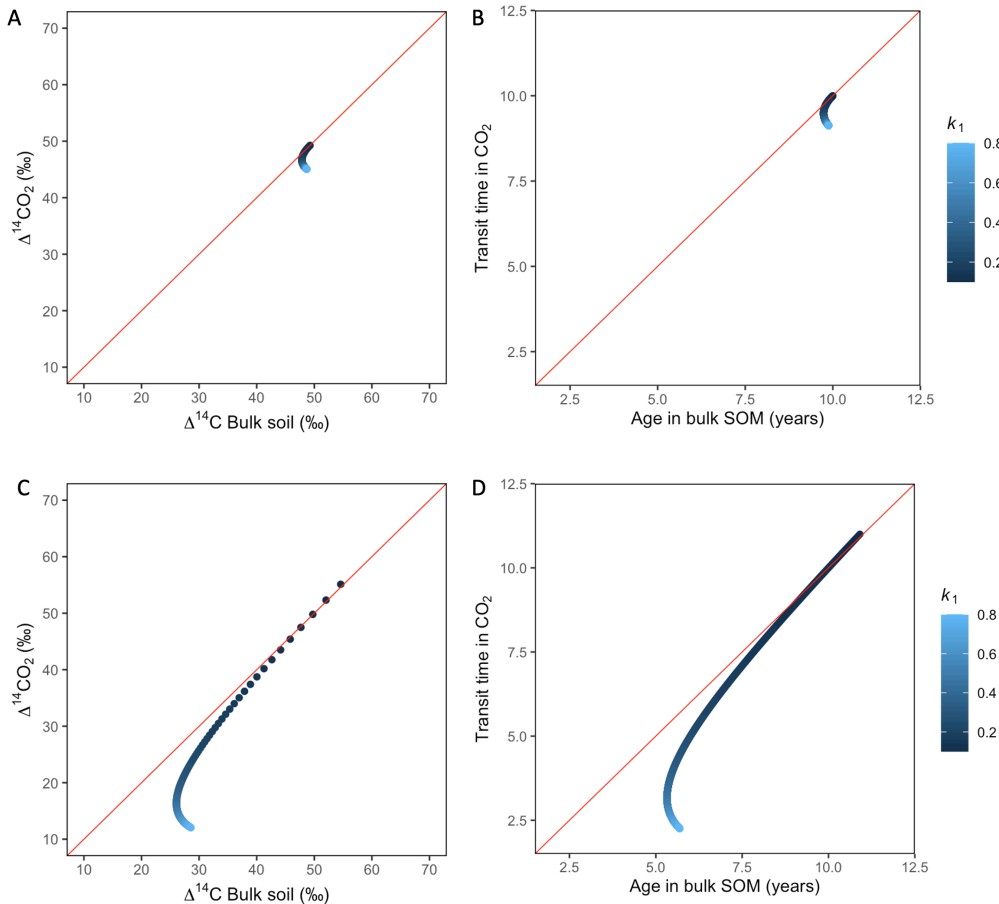

**Figure A2.** Predictions of $\Delta^{14}C$ in bulk soil vs $\Delta^{14}CO_2$ with their equivalent simulation of mean age in bulk soil vs mean transit time in $CO_2$ for parallel (panel A and B) and series model structure (C and D). Variation of $k_1$ with $\alpha = 0.1$ and $\gamma = 0.1$.




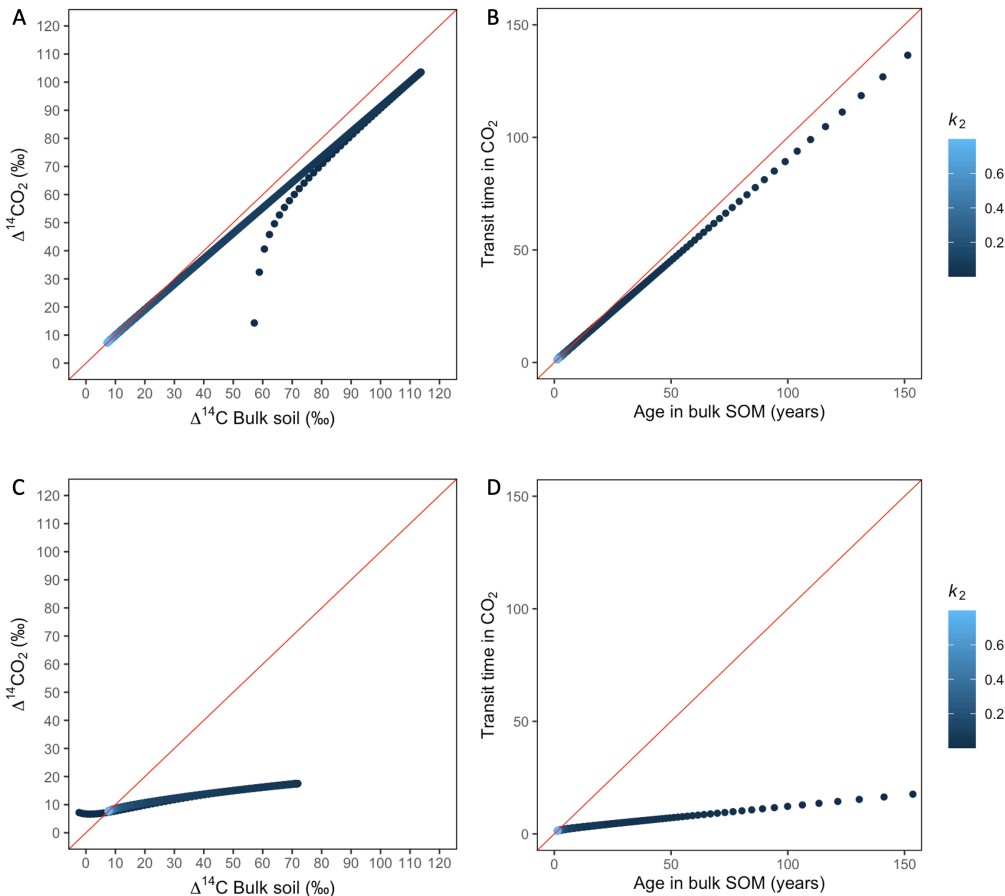

**Figure A3.** Predictions of $\Delta^{14}C$ in bulk soil vs $\Delta^{14}CO_2$ with their equivalent simulation of mean age in bulk soil vs mean transit time in $CO_2$ for parallel (panel A and B) and series model structure (C and D). Variation of $k_2$ with $\alpha = 0.1$ and $\gamma = 0.1$.





## - $\Delta^{14}C$ variation in slow cycling systems

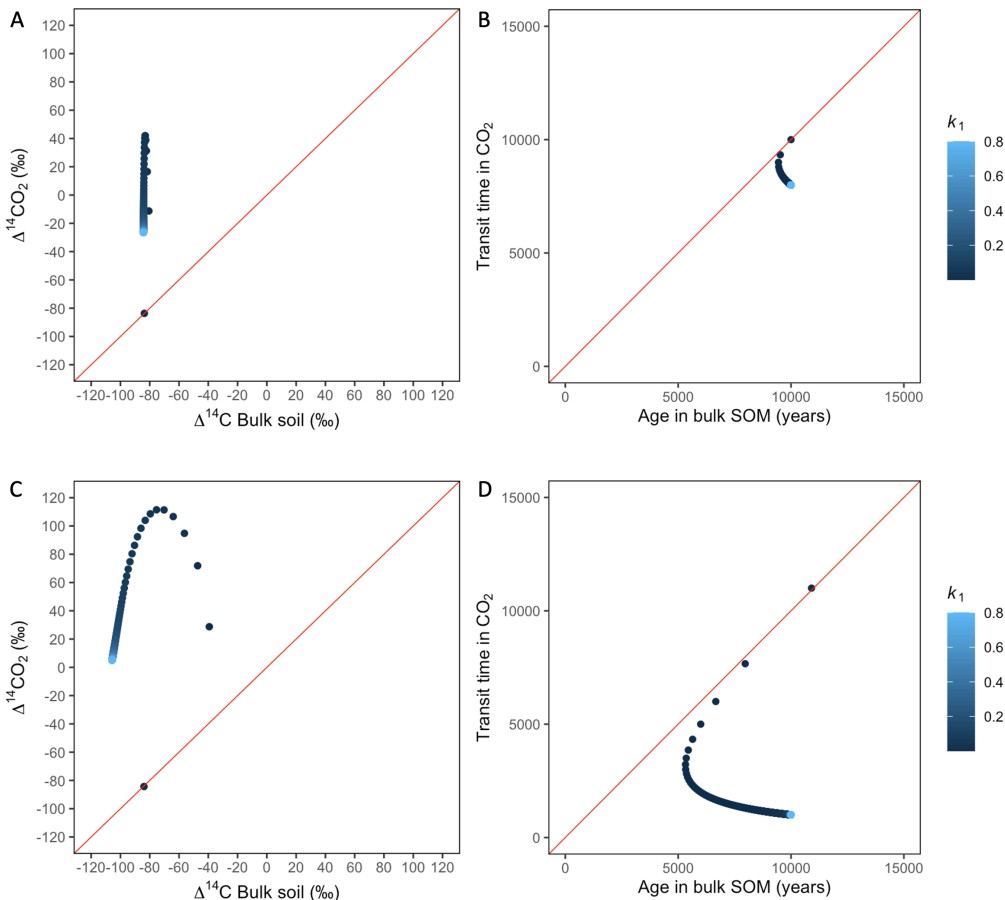

**Figure A4.** Predictions of $\Delta^{14}C$ in bulk soil vs $\Delta^{14}CO_2$ with their equivalent simulation of mean age in bulk soil vs mean transit time in $CO_2$ for parallel (panel A and B) and series model structure (C and D). Variation of $k_1$ with $\alpha = 0.1$ and $\gamma = 0.2$.





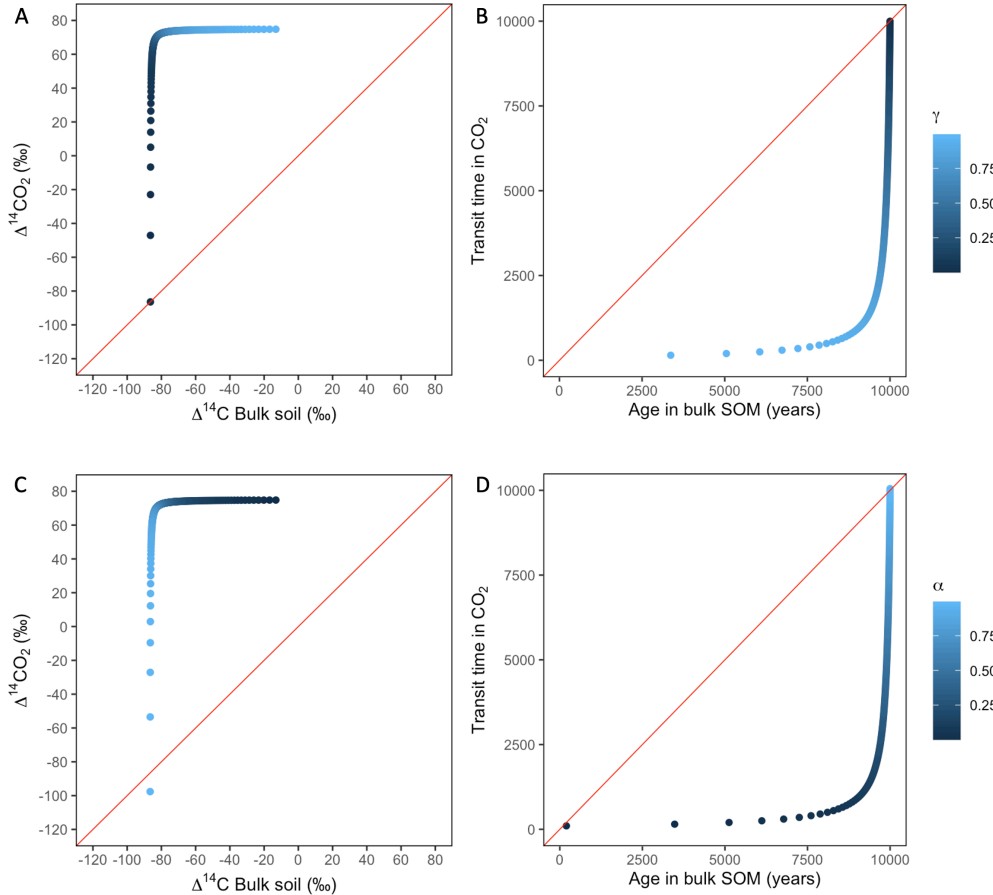

**Figure A5.** Predictions of $\Delta^{14}C$ in bulk soil vs $\Delta^{14}CO_2$ with their equivalent simulation of mean age in bulk soil vs mean transit time in $CO_2$ for parallel (panel A and B) and series model structure (C and D). $k_1 = 0.01$, $k_2 = 0.0001$, with $\alpha = 0 - 1$ and $\gamma = 0 - 1$.



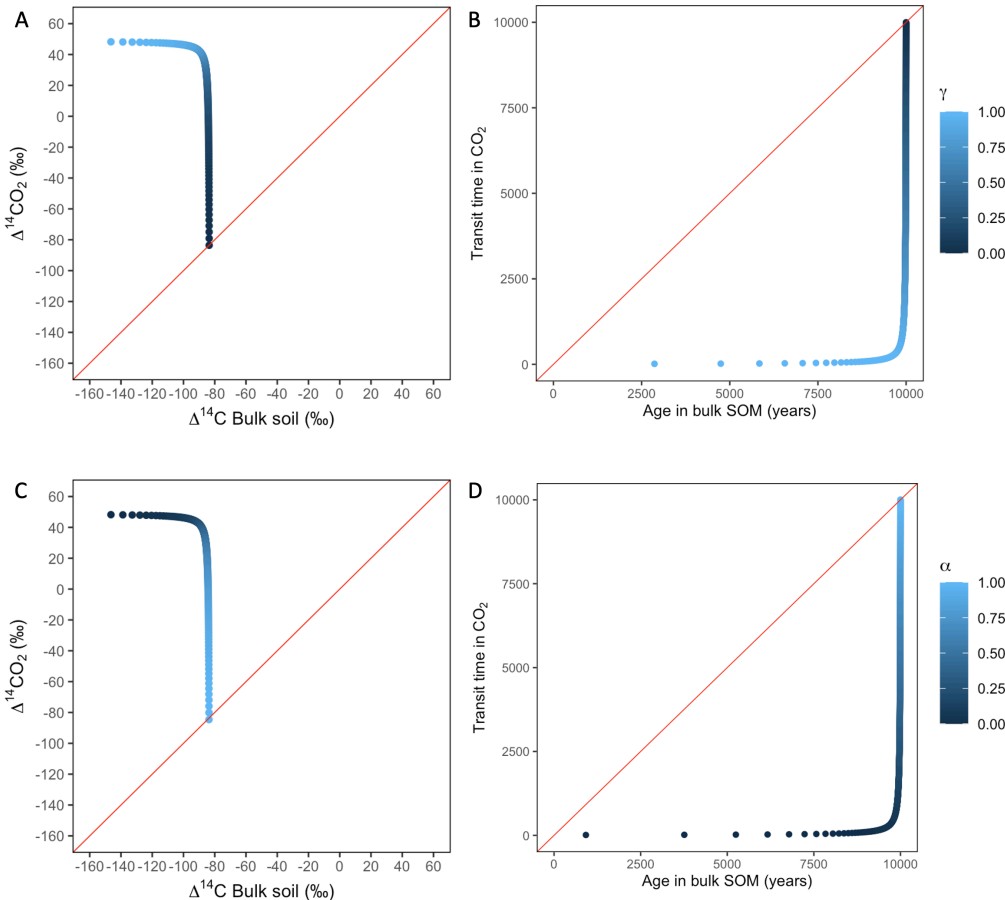

**Figure A6.** Predictions of $\Delta^{14}C$ in bulk soil vs $\Delta^{14}CO_2$ with their equivalent simulation of mean age in bulk soil vs mean transit time in $CO_2$ for parallel (panel A and B) and series model structure (C and D). $k_1 = 0.1$, $k_2 = 0.0001$, with $\alpha = 0$ - 1 and $\gamma = 0$ - 1.



**- $\Delta^{14}$C curves for bulk soil, respired CO$_2$ and atmosphere for a two-pool system in parallel and series structure**

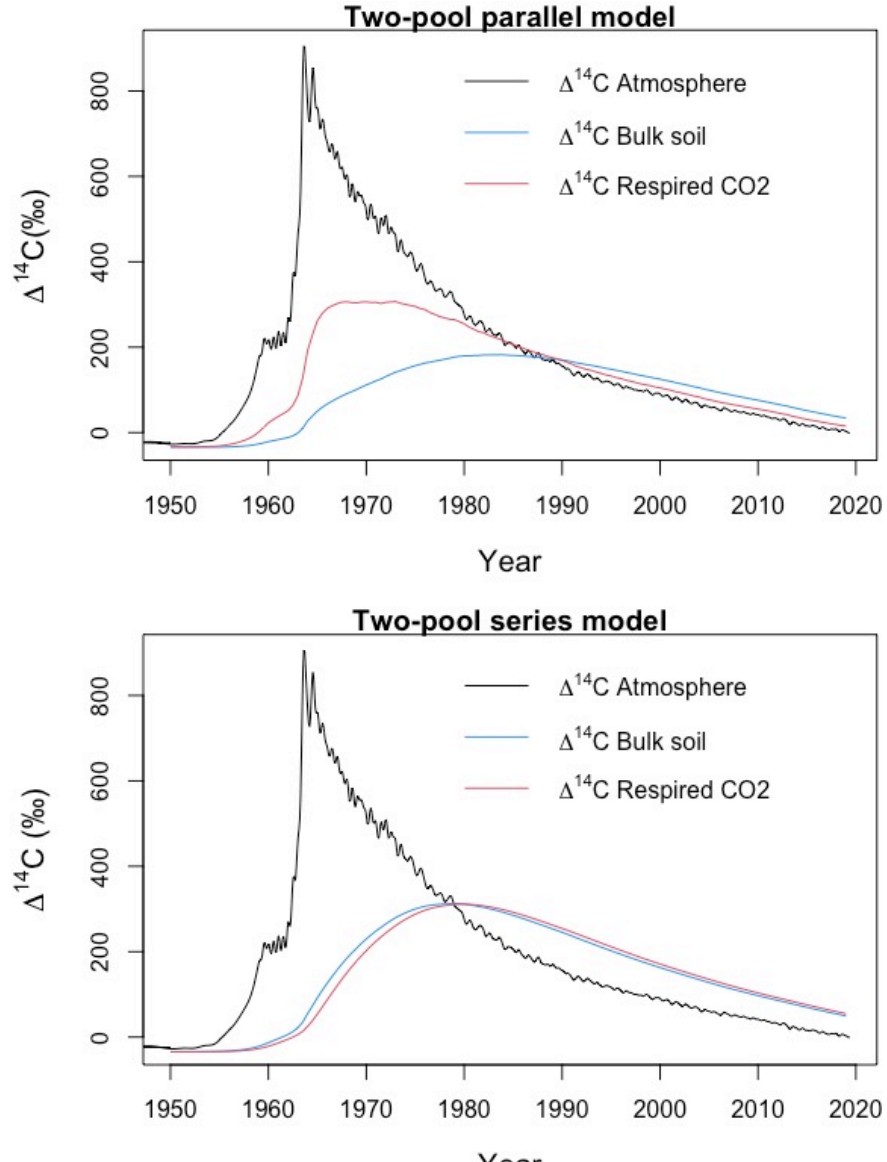

**Figure A7.** Variation of $\Delta^{14}$C values in atmosphere, bulk soil and respired CO$_2$ for a two-pool soil during the period 1950 - 2019. The similarity between $\Delta^{14}$C values may indicate both a well-mixed homogeneous system (type I) or a retention system (type II) where the bulk soil and the respired flux record the same value. Nonetheless, the mean transit time for the bulk and the soil respiration might be different due to the contrasting decomposition rates of the fast ($k_1 = 0.8$) and the slow pool ($k_2 = 0.1$). Simulation conducted with $\alpha = 0.8$ and $\gamma = 0.8$.



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
