# Peer review of "Moisture and temperature effects on the radiocarbon signature of respired carbon dioxide to assess stability of soil carbon in the Tibetan Plateau"

_EGUsphere, 2023_

## Author Comment (AC1)

**RC1**: 'Comment on egusphere-2023-210', Anonymous Referee #1, 23 Mar 2023

We thank referee # 1 for his/her constructive comments which were highly appropriate and contributed significantly to the improvement of our manuscript. Here we quote his/her comments in *italics* and provide our answers below each of them.

- *Tangarife-Escobar et al. present a well-executed and thoughtful experiment carried out in an important geographic region. I enjoyed reading the work, and find it well argued and well composed for the most part. I would suggest that some additional data be included in the main text (which is now not given at all, or is relegated to the supplementary/appendix materials) to help clarify and strengthen arguments in the discussion. The modeling and empirical components are a bit disconnected from one another as currently presented. If the experimental results cannot be used directly in the soilR simulations, perhaps some of the data may be presented in merged figures to help the reader more directly understand the connections between these two components of the work (see detailed comments below).*

  We carefully addressed this issue to make sure that the connection between the experimental and the modelling approaches is clearer. Please see comments below.

  *Detailed comments:*

- *Abstract line 17: The statement that temperature is a significant variable contradicts the results stated earlier in the abstract.*

  We reinterpreted this assertion and corrected as follows:

  "We conclude that the stability of carbon in the peatland and grassland soils of the QTP depends strongly on the direction of change in moisture and how it affects the rates of SOM decomposition while temperature regulates the amount of fluxes."

  Additionally, although temperature did not affect the $\Delta^{14}C$ values from bulk soil or $CO_2$ in the incubations, it does modulate the stability of carbon through changes in decomposition rates (we added Figure 5 C-D which clearly shows this pattern). The latter can be seen clearly in the results of the simulations, where the decomposition rates ($k$) affect the $\Delta^{14}C$ in both slow and fast cycling systems.

- *Lines 56-64: This is a very clear and succinct explanation of C pools and turnover. Nicely done!*

- *Line 110: How did the presence of inorganic C potentially affect the Delta 14CO2 values?*

  According to our calculations, the reported percentages of inorganic carbon of 0.05 and 0.06 as found in the analyzed peatlands and grassland soils, respectively, would affect the $\Delta^{14}C$ in less than 0.05 ‰, which we consider is negligible for the purpose of our study.

- *Methods: Soil incubation times… the duration of the incubations is cited many times in the discussion as a potential confounding variable in the interpretation of the Delta 14CO2 data. However, the length of the incubations is not given in the methods. The length of the incubations should be added to the manuscript along with a discussion of how variable lengths of incubations for the individual treatments might have influenced*

*the Delta 14CO2 data. I'm assuming different treatments were incubated for different lengths of time since the methods indicate that they were incubated until a certain amount of CO2 was produced, and given the differences in respiration rates given in the appendix the length of incubation time might have varied by an order of magnitude? Could this have an influence on the age of C being respired (i.e. longer incubation times allowed for decomposition of more structurally and/or chemically "stabilized" substrates)?.*

The duration of the incubation is mentioned in the methodology (section 2.2). However, we added a column with the specific time of incubation for every treatment in the table A1 and reference it in section 2.2. We also corrected the mean $CO_2$ respiration values which were interchanged between peatland and grassland. As we emphasized in the discussion, the incubation duration could indeed have an effect on the type of C respired from the soil and therefore on the transit time of the respired $CO_2$ since "younger" SOM might have been oxidized faster than more stable "older" SOM.

We added:

"An important aspect that has been observed in soil incubations, is that $CO_2$ accumulation decreases or even stops after a certain period probably due to the $CO_2$ saturation of the limited headspace in the incubation flasks, which depends on the SOC content. From this we could deduce that a higher respiration rate is followed by fast saturation of the headspace in soils with high SOC and therefore only the carbon firstly decomposed (usually labile) will be present in the $CO_2$. For example, Azizi-Rad et al. (2022) found the respiration rate to decline after 14 days holding the soil at 10 C. In this sense, incubations with high TOC may run out of headspace soon and hinder the respiration of old carbon that requires longer times to be decomposed"

Complementarily, we have observed in recent incubations (Tangarife et al, in preparation) that the $CO_2$ respiration from soil stops after few days probably due to the saturation of the limited headspace in the incubation flasks. From this we could deduce that the isotopic signature of soil respiration can only be properly measured if the $CO_2$ accumulation is guaranteed.

Modified Table A1.

**Table A1.** Mean daily $CO_2$ respiration for incubated soils under temperature and WFPS variation.

| Ecosystem | Temperature (°C) | WFPS (%) | Mean $CO_2$ respiration (mg $CO_2$ g soil $^{-1}$ day $^{-1}$) | $\sigma$ | Incubation time (days) n = number of samples |
|---|---|---|---|---|---|
| Grassland | 20 | 95 | 0.040 | 0.035 | 2n = 32, 1n = 19 |
| | 20 | 60 | 0.028 | 0.016 | 4n = 19 |
| | 15 | 95 | 0.024 | 0.008 | 3n = 30, 1n = 15 |
| | 15 | 60 | 0.016 | 0.004 | 4n = 30 |
| | 10 | 95 | 0.011 | 0.011 | 3n = 66 |
| | 10 | 60 | 0.008 | 0.023 | 6n = 66 |
| | 5 | 95 | 0.005 | 0.003 | 5n = 67 |
| | 5 | 60 | 0.004 | 0.001 | 4n = 67 |
| Peatland | 20 | 95 | 0.510 | 0.001 | 5n = 13 |
| | 20 | 60 | 0.371 | 0.003 | 4n = 13 |
| | 15 | 95 | 0.330 | 0.004 | 5n = 13 |
| | 15 | 60 | 0.234 | 0.003 | 3n = 13 |
| | 10 | 95 | 0.209 | 0.002 | 5n = 13 |
| | 10 | 60 | 0.134 | 0.001 | 5n = 13 |
| | 5 | 95 | 0.066 | 0.000 | 4n = 13 |
| | 5 | 60 | 0.044 | 0.001 | 5n = 13 |

- *Methods: What is the reasoning behind the choice of WFPS values? Ninety-five percent is very high. Doesn't this value inhibit evolution of gases from the soil matrix? How/why were 65% and 95% chosen?*

We choose 60 and 95% WFPS because of several reasons. First, to compare our results to Sierra et al. (2017) and Azizi-Rad et al. (2022) who had conducted previous incubation experiments on sensitivity of soil respiration to temperature and moisture. Azizi-Rad et al. (2022) analyzed the same grassland soil of our study and found that temperature was the factor limiting soil decomposition provided that moisture and oxygen were sufficiently available. Hence, we were aware that our WFPS levels must contain enough water and oxygen for ensuring soil respiration. Finally, one of the main objectives of our study was to observe the behavior of $\Delta^{14}C$ and transit times of C in soils under degradation processes such as from high saturation towards drying in peatland soils and from seasonally frozen to water saturated in grasslands.

We added the next sentences to the section 2.2 in methodology:

"Each of the sets was placed at two different WFPS levels (60 and 95 %) which were selected in order to reassemble the thaw and consequent water saturation of seasonally frozen soils in grasslands; and for peatland soils, the process of drying (through artificial desiccation) after high water saturation"

On the other hand, 95% of WFPS is certainly a high level of moisture saturation, nonetheless, the incubated soil had still wide contact with oxygen, allowing SOM oxidation and $CO_2$ accumulation.

- *Table 2: Unclear what is being compared here. Is the anova between grasslands and peatlands at each treatment level of temp/moisture? Or is it comparing different levels of temp within each soil category? It's confusing because the soils weren't radiocarbon dated \*after\* incubation, correct?*

We apologize for the misunderstanding. We are comparing here the variation of $\Delta^{14}C$ values of bulk soil and $CO_2$ for each type of ecosystem with the objective to evaluate the effect of temperature, WFPS and WFPS x temperature treatments. $\Delta^{14}C$ values from bulk and $CO_2$ were obtained after the incubations. We reformulated the caption of Table 2 as follows:

"Summary of *p*-values obtained from ANOVA tests for the $\Delta^{14}C$ of bulk soil and the $\Delta^{14}CO_2$ of grassland and peatland soils after incubation. *p*-values are given for the independent effect of temperature (T) and WFPS as well as the integrated effect of temperature and WFPS (T • WFPS)."

And also added a sentence in the methodology to clarify:

"Radiocarbon analysis were conducted in the bulk soil of each sample after the incubation."

- *Table 3: This is really a lot of different conditions... and on top of that you discuss the type I, type II or type III systems. How do these three things relate to one another ("fast/slow", "parallel/series", "type I/II/II")? Also, please add to the "System" column "grassland" and "peatland" in addition to "fast" and "slow". I know it's in the text directly below, but it would help the reader keep on top of all the modeling approaches.*

The classification in type of system (I, II and III) indicates the relationship between age and transit time. We established the possible type of systems for specific ecosystems and environmental conditions based on the $\Delta^{14}C$ values obtained from the incubations. Additionally, from the modelling exercise we observed how the soils moved along these types of system when modifying the internal characteristics and the model structure.

We added a sentence to clarify this relationship:

"Our simulations mimicked a fast cycling grassland and a slow cycling peatland by differentiating the ranges of decomposition rates (Table 3) and showed how the modelled conditions affected the type of system (I, II and III).

We would prefer not adding "grassland" and "peatland" to the "System" column in Table 3. Our modelling approach reassembles the conditions for slow and fast cycling systems and targets the $\Delta^{14}C$ values found in our incubations. As we do not fit the models to any observed values from the incubated grassland and peatland soils but rather explore which ranges of conditions would be useful to describe the $\Delta^{14}C$ values, we decided not to indicate in the table the names "grassland" and "peatland" to not to mislead the reader. Instead, we clarified in caption of Table 3 that the parameters used for the simulations might be helpful to understand $\Delta^{14}C$ values as well as age and transit time in grasslands and peatlands.

"Range of parameters used for simulations in a SOC decomposition model for fast and slow cycling systems (Tangarife-Escobar et al. 2023). These ranges explore the

required conditions to describe the $\Delta^{14}C$ variation found in the incubated grassland and peatland soils and their equivalent age and transit time relationship. The target $\Delta^{14}C$ intervals are shown as boxes in the Figures 6 to 9."

Additionally, to establish a more accurate connection between the $\Delta^{14}C$ values from the experiments and the simulations, we added some panels indicating the target region on the Figure 6-9 and A1 to A6 (see comment below).

[Figure]

- *Figure 5: I feel that it is important to have an additional two panels in this figure showing the total amount of C respired by each of the treatments for a given length of time or the respiration rates. This information is referenced in the discussion, but I don't see it anywhere. In the discussion, the manuscript makes a point about the relative importance of the age vs. the amount of respired C, so the amounts should be shown. See additional comment regarding appendix table A1 below.*

This is an excellent suggestion. We added panels with the respiration rates, calculated over the total length of the incubation for each treatment. The times of incubation are specified in table A1. Here the correlation between ecosystem, temperature, WFPS and $CO_2$ respiration rates becomes more evident. In that sense, we modified Figure 5 and its caption, see comment below where Figure 5 is also addressed.

Additionally, we presented the results on respiration rates more in detail in section 3.1:

"Higher temperature and WFPS caused an increase of $CO_2$ fluxes from respiration in the treated incubated soils (Table A1, Fig. 5 C-D). In both ecosystems, wetter conditions showed higher respiration rates and higher slopes as the temperature increased. The absolute amounts of $CO_2$ produced from peatland soil was in average 14 times higher than from grassland soils for every independent treatment."

- *Figures 4 and 5: The 10 deg C thing… something unique seems to be happening at this temperature in the peatland soils. Do you have some explanatory hypotheses? This temperature also has strong outliers in both soil types (Figure 4), would you please comment on this?*

  The appearance of outliers in the 10 °C treatments for both of the ecosystems was one of the main questions emerging from our experiments. After discarding any measurement issue, we conducted the simulations to evaluate if such values belonged to the range in which $\Delta^{14}C$ moved for specific soil characteristics. However, the $\Delta^{14}C$ curves obtained from the simulations did not describe the behavior of the outliers.

  We elaborated on the outliers explanation as follows:

  "Finally, although extremely depleted $\Delta^{14}C$ values are rare, they are probable since the measured carbon particles represent one value, which can fall on the tail of the system-age probability distribution. We interpret that outlier results belong to carbon particles that have remained for very long time in the system (out of the mean values) whose specific soil characteristics were not captured in our model structure or internal soil conditions. The reasons behind the occurrence of these outliers at the specific 10 °C remain to be investigated."

- *Figures 6-9: Please label all these panels of figures as "grass vs peat" and "fast vs slow". Preferably in the figure itself, but at least in the caption. This will help the reader more easily keep track of what they're looking at.*

  Every figure from 6 to 9 corresponds to only one set of conditions. For example, Figure 6 shows the simulation for a fast cycling system, which targets the $\Delta^{14}C$ values of the incubated grassland soil. Therefore, we added this information in the caption of each figure (Fig. 6-9 and Fig. A1-A6) which were accordingly adapted as follows:

  "Predictions of $\Delta^{14}C$ in bulk soil vs $\Delta^{14}CO_2$ with their equivalent simulation of mean age in bulk soil vs mean transit time in $CO_2$ for parallel (panel A and B) and series model structure (C and D) for a fast cycling system. $k_1 = 0.8$, $k_2 = 0.1$, with α = 0 - 1 and $\gamma$ = 0 - 1. Green box represents the range of measured $\Delta^{14}C$ values obtained from incubated grassland soils for bulk (20 to 74 ‰) and $CO_2$ (10 to 85 ‰), excluding outliers."

  "Predictions of $\Delta^{14}C$ in bulk soil vs $\Delta^{14}CO_2$ with their equivalent simulation of mean age in bulk soil vs mean transit time in $CO_2$ for parallel (panel A and B) and series model structure (C and D) for a slow cycling system. Variation of $k_1$ with α = 0.1 and $\gamma$ = 0.2. Brown box represents the range of measured $\Delta^{14}C$ values obtained from incubated peatland soils for bulk (-90 to -65 ‰) and $CO_2$ (-18 to 25 ‰), excluding outliers."

- *Model/data fusion: Can soilR not use the Delta 14C of the respired C to constrain alpha/gamma and k1/k2 values? Or is that too computationally intensive at this point? I find the paper to be well written, but there is not a lot of integration of the incubation data with the modeling exercise. What would help me understand the connections would be plotting (some? All?) the data from figure 4 onto figures 6-9. This would directly show me how the experimental results map onto the different modeled scenarios. This would really the reader more quickly understand the connections between the type I/II/III systems and the model parameters (gamma/alpha and decomposition rate constants).*

Please see comment above. Our objective was not to fit a model to the observed incubated data but rather exploring how certain set of conditions (internal characteristics of soil) related to SOM decomposition rates affected the $\Delta^{14}C$ values and their corresponding mean age and mean transit time (as added in section 2.3). Our experimental results provided radiocarbon data with relatively large variance, which is challenging to model in a classical, parameter optimization, sense. Nonetheless, the $\Delta^{14}C$ data in the bulk soil and respiration obtained from the incubations were used as the target x-y-coordinate-space of our simulations. To show this connection in a more precise way we modified the captions and added boxes to the simulation figures as explained in the previous comments. We hope that in this way, the link between the experimental and the modelling sections of our manuscript becomes more tangible.

- *Table A1: Ok. Here is where the significant temperature effect is... this looks like a pretty linear response of respiration rate to increasing temperatures. Why not include this result in the main text? I think it's alluded to in the abstract, but there's no actual evidence of a temperature effect included in the main text at this point (maybe add to figure 5).*

[Figure]

This is a very appropriate idea. We added panel C and D to figure 5 to illustrate the response of mean respiration rates to temperature and WFPS. In consequence we modified Figure 5 and its caption as follows:

[Figure]

"Figure 5: Comparison between $\Delta^{14}C$ of respiration from incubated grassland (A) and peatland (B) soil at different temperature levels under WFPS = 60 and 95 %. Black points represent minimum and maximum values out of the range between quartile 1 and 3 (25 to 75 % of the data). The quartile 50 (median) represented by the line inside the box indicates the midpoint value in the frequency distribution. Box for the treatment WFPS= 60 % and T= 10 °C shows a large dispersion of the 50 % of the data, which is explained by the outliers observed in Fig. 4. Additional panels indicate the respiration rates (mg $CO_2$ • g soil -1 • day -1) for each treatment based on the total duration of incubation (see Table. A1). Response of mean respiration rates to temperature and WFPS treatments in grassland (C) and peatland (D) soils."

And added the next lines in the text:

"Higher temperature and WFPS caused an increase of $CO_2$ fluxes from respiration in the treated incubated soils (Table A1, Fig. 5 C-D). In both ecosystems, wetter conditions showed higher respiration rates and higher slopes as the temperature increased. The absolute amounts of $CO_2$ produced from peatland soil was in average 14 times higher than from grassland soils for every independent treatment."

- *Conclusions: The conclusions lack punch. What are the broader implications of this work? How do the experimental treatments relate to current climate projects for the QTP? Will the peatlands dry out and change this from a type X to a type X system? Will the grasslands get hotter and therefore respire X more gigatons of C on an annual basis? The introduction states that these soils are being studied because of they hold*

*vast stores of C. What do your experiments suggest for the fate of these C stores under future climate scenarios? .*

We complemented and modified the conclusions as follows:

"From our modelling approach, we conclude that radiocarbon can be used as a tool to understand SOM persistence through the use of the concepts of mean age and mean transit time and their mutual relation. Our simulations were able to reassemble the $\Delta^{14}$C values obtained from incubations and showed that modelled $\Delta^{14}$C values differed widely between slow cycling systems and fast cycling systems. We found that low values of decomposition rates, more common in slow cycling systems, modified the behavior of $\Delta^{14}$C patterns due to the incorporation of $^{14}$C-bomb in the soil system. Hence, the correspondence between these mutual relations strongly depended on the internal dynamics of the soil and its interaction with the environment. For this reason, the acquisition of empirical data from soils (number of pools, I, C, $k$, $\gamma$ and $\alpha$) along with the correct setting of model structure will improve our understanding on the stability of carbon in the soils of a changing QTP. In this way, current changes in climate patterns and land cover alteration may have a larger impact on the Zoige peatlands than on the grasslands given the vulnerability of large carbon stocks to be destabilized by changes in temperature. Nevertheless, the interaction with moisture may dampen or amplify the temperature effect, adding uncertainty on the future trajectories of soil carbon in the Qinghai-Tibetan Plateau."

New references:

Sierra, C. A., Malghani, S., and Loescher, H. W.: Interactions among temperature, moisture, and oxygen concentrations in controlling decomposition rates in a boreal forest soil, Biogeosciences, 14, 703–710, 2017a.

---

## Author Comment (AC2)

**RC2**: 'Comment on egusphere-2023-210', Anonymous Referee #2, 25 Jul 2023

We would like to thank referee # 2 for providing a review on our manuscript. All the comments were thoroughly addressed and complemented with the replies to the first referee. The main comments of his/her review are provided below in *italics*, with our reply in normal font.

- *In this study, Tangarife-Escobar and co-authors incubated peatland and grassland soils at four different temperatures and two water-filled pore spaces to better understand temperature and moisture effects on the 14C signature of bulk and respired soil carbon. They also used a mathematical model to analyze how decomposition rates and other soil parameters affect the $\Delta$14C of bulk and respired C and their relationship. Papers that investigate the relationship between carbon age and transit time are crucial in order to better understand carbon destabilization with changing environmental conditions. Although this paper has the potential to be impactful, it needs to be revised to address important flaws in the interpretation of the incubation results and to improve the link between the incubation and mathematical model.*

- *Overall, there are major issues with the interpretation of the incubation results and this is my main concern about this paper. The authors stated that temperature and soil manipulations caused a response in the direction of the $\Delta$14C (Abstract and Lines 233-234); however, these statements are not supported by the statistical analysis, which show that temperature had no effect on the $\Delta$14C of bulk soil or respiration. The authors do not present any information on the statistical model that was used to interpret these results in the methods section, and as such, it becomes difficult to understand how they arrive at some conclusions related to the $\Delta$14C and CO2 flux (e.g., was site included in the stats model? What was the stats model for the CO2 flux?). Additionally, the text in the results section rarely includes wording on whether a main effect was significant, and sometimes the authors will discuss the interactive effect of moisture and temperature on $\Delta$14C, even though the interaction was not significant.*

  We apologize for the misunderstanding. Certainly, temperature did not show any effect on the $\Delta^{14}C$ values, but rather on the magnitude of fluxes, which directly affects the release of "old" carbon if the moisture conditions allow it. To illustrate this, we added Figure 5 (C-D) which provide a clearer picture on the relationship between $CO_2$ fluxes and treatments and we indicate the respiration rates for each treatment inside small panels. It is relevant to point out that although our experiment did not find significant evidence of a correlation between temperature variation and $\Delta^{14}C$ values as assessed by a two-way ANOVA (type III), other authors have found that temperature is the main driver of SOC decomposition. Hence, we clarified in the abstract and along the manuscript that the influence of the temperature occurs only on the fluxes.

  Additionally, we added information on the statistical model we used for the interpretation (see comment below).

  Regarding the measurement of the $CO_2$ fluxes, we added a paragraph in the methodology explaining its calculation and added Figure 5 (C-D) to provide a deeper interpretation on $CO_2$ fluxes patterns at different treatments. A more detailed answer can be seen in "reply to referee #1".

  Finally, the interaction effect of temperature and soil moisture manipulations on $\Delta^{14}C$ values was evaluated from the perspective of the incubation results and no significant

difference was found by the two-way ANOVA (type III), which is appropriate for unbalanced data since our results did not have equal number of values (see comment on section 2.3). Nonetheless, we argue based on our observations of temperature variation effect on $CO_2$ fluxes and the studies of other authors, that temperature and soil moisture strongly influence the decomposition of SOM and therefore would have an influence on $\Delta^{14}C$ values, age and transit time. $\Delta^{14}C$, as opposed to $CO_2$ fluxes, is a variable that may show more complex dynamics due to the trends of the atmospheric bomb curve, motivating our subsequent modelling analysis.

- *It is also unclear how the incubation informed the SOC decomposition model, for example did the CO2 flux response to temperature and moisture get used in the model to influence the decomposition constants, or were these based on Manzoni et al. 2009 (Lines 182-185)? If the incubation did not influence the model, then what was the reason to include the incubation results in the paper? The authors should expand on the link between the incubation and model and how they influenced each other.*

As previously mentioned, $\Delta^{14}C$ is a variable that can show much more complex behaviors than $CO_2$ fluxes because of the dynamics of the atmospheric bomb curve, which can potentially lead to non-statistical differences among treatments even though decomposition rates and transit times may respond strongly to changes in temperature and moisture.

We further elaborated on the reasons to use a mathematical model and how it relates to the incubation data (section 3.2 and comments to the first referee). The $CO_2$ results were not used explicitly in the mathematical model since the objective was to explore behaviors in the dynamics of $\Delta^{14}C$ as decomposition rates change, and how these behaviours differ from those of age and transit time. Because we aimed at exploring a wide range of values of decomposition rates, we didn't use the specific results of the incubation experiment, but we rather tried to explore the directions in which $\Delta^{14}C$ would change as decomposition rates change. The incubation results were useful to show the type of storage system of each of the ecosystems as well as to gain knowledge on the influence of different treatments on the $\Delta^{14}C$ values. The lack of clear response to these treatments observed in Fig. 4 along with the presence of outliers were approached by the model simulation, which contrasted slow and fast cycling systems to observe what other additional parameters could modulate the behavior of $\Delta^{14}C$ values along with the age and transit time metrics. Specifically, the $\Delta^{14}C$ data obtained from the incubations were used as the target space of our simulations, for what we added some panels on the Figures 6 to 9 to indicate the aimed $\Delta^{14}C$ range. By doing that, we hope the connection between the incubations and the modelling results be more substantial.

- *Finally, the writing can be improved and the authors should check the manuscript for typos and inconsistency in terminology.*

We carefully addressed this comment by proofreading and adjusting terminology consistently along the manuscript.

*Detailed comments:*
*Abstract*

- *The abstract does not currently satisfactorily connect the results to the big takeaway statements and implications for the net carbon balance of the area.*

We elaborated on the implications of our results for the ecosystems future and stated them in the abstract and conclusions.

- *Line 10: Is this bulk soil 14C?*

  We clarified the sentence as follows:

  "From our incubations, we found that $^{14}$C values in bulk and $CO_2$ from peatland were significantly more depleted (old) than from grassland soil."

- *Line 14: Consider not using terms like k in the discussion of results, but instead refer to the decomposition rates. To a general audience saying what the effect of a 'low k value' may not carry a lot of meaning; however, discussing the impacts of 'low or high decomposition rates' will aid in the interpretability of the results.*

  We changed "k" for "decomposition rates" in the discussion when appropriate, to help the readability and interpretation of results.

- *Lines 13-16: It is not clear how these two sentences are linked, or how the first statement leads to the second. It seems like they should be switched: The correspondence between ∆14C and age and transit time strongly depended on the internal dynamics of the soil (k, α, γ and number of pools) as well as on model structure. When decomposition rates were low (low k values), the (replace "modified ∆14C" by the direction in which the ∆14C changed, did it increased or decreased the age of bulk/respired CO2?) due to the incorporation of 14C-bomb in soil (does the incorporation of 14C-bomb mean anything to the reader at this point in the abstract? Consider writing what this means (e.g., the proportion of C cycling on decadal timescales increased). What does this result mean for carbon cycling in wet/dry or cold/wet systems? The abstract is missing the implication of the results.*

  We reformulated these sentences for a better connection between them and added the specific implications of our results as follows:

  "In our models, the correspondence between ∆14C, age and transit time highly depended on the internal dynamics of the soil (*k, α, γ* and number of pools) as well as on model structure. We observed large differences between slow and fast cycling systems, where low values of decomposition rates modified the $\Delta^{14}$C values in a non-linear pattern due to the incorporation of modern carbon ($^{14}$C-bomb) in the soil. We conclude that the stability of carbon in the peatland and grassland soils of the QTP depends strongly on the direction of change in moisture and how it affects the rates of SOM decomposition while temperature regulates the amount of fluxes. Current land cover modification (desiccation) in Zoige peatlands and climate change occurring on the QTP, might largely increase $CO_2$ fluxes along with the release of old carbon to the atmosphere potentially shifting carbon sinks into sources."

- *Lines 16-18: Why are the authors concluding that the stability of carbon depends strongly on changes in temperature if this did not affect the ∆14C of respired CO2 in either soil?*

  We reinterpreted this assertion and modified accordingly along the manuscript (see previous comment).

- *Lines 18-19: It is not clear how modeling improved predictions on interactions between terrestrial and atmospheric carbon.*

We modified the end of the abstract to highlight the implications of our results (see comment above).

*Introduction*

- *The introduction is a bit choppy. Some of the paragraphs can be combined and the order should be reconsidered to improve the readability of the paper. For example, paragraphs 3 and 6 are both about carbon stabilization/de-stabilization, yet they are broken up by paragraphs 4 and 5.*

As paragraph 3 introduces the concept of stabilization mechanisms and their relation to SOM persistence and paragraph 6 explains how $^{14}$C can be altered by stabilization and destabilization mechanisms, we consider that a bridge (paragraph 4 and 5) that introduces both age and transit time and how they are studied through radiocarbon is necessary to keep the logic sequence of the introduction.

- *Please be consistent with word choices, e.g., de-stabilization, destabilization, (de)-stabilization, and the use of carbon or 'C.'*

We changed to the terms "destabilization" and "carbon" consistently throughout the entire manuscript.

- *Please add text in the introduction on how the nuclear weapons test enriched atmospheric 14C to aid in the interpretation of positive vs negative values.*

Thanks for the recommendation. We added some sentences to explain the relationship between nuclear weapons and $\Delta^{14}$C values.

- *Line 4: add comma for non-essential clause: ", and in consequence,"*

We corrected along the entire manuscript as suggested.

- *Line 34: Comma before non-essential clause ", which store"*

We corrected along the entire manuscript as suggested.

- *Line 37: Driving the net carbon balance to become a C source? Is it already a source of C?*

We changed the verb "driving" to "modulating" since the sense of the sentence is to indicate that according to the referenced authors, climate change and land cover change are affecting the carbon balance through effluxes. The direction of such change from sink to source or vice versa differs across different studies and sub-regions of the QTP.

- *Line 80-81: Add Pegoraro et al. 2021 to the list of citations for release of old C from deep soil layer after drainage.*

Very useful recommendation. We added as suggested.

- *Lines 81-82: Is this a perennial or seasonal frozen layer?*

We modified the sentence to clarify as follows:

"Such a process might be occurring in the Zoige peatland soils due to the presence of seasonal frozen layers (Liu et al., 2021; Yang et al., 2022)."

- *Line 86: What is the impetus for the older C increase with increasing moisture? One would think that higher soil moisture would decrease the decomposition of SOC, and thus preserve older C, based on the citations in the previous paragraph.*

This is a crucial observation whose reasoning relies on the starting moisture point of the soil. According to Sierra et al. (2017) and Azizi-Rad et al. (2022), SOM decomposition is limited at extreme values of soil moisture and oxygen levels (0 % or 100 %). So, as long as oxygen and moisture are available (as in our 95 % WFPS level), SOM decomposition and $CO_2$ release (old in the case of the peatlands) would occur, while if the moisture decreases to the extreme, decomposition would be highly restricted independently of temperature and oxygen levels.

Therefore, we rephrased the hypothesis as follows:

"Changes in temperature and moisture contribute to the destabilization of carbon in soils from the QTP. Hence, we hypothesize that higher temperature would increase the age of respired $CO_2$ as well as changes in soil moisture would increase or decrease (depending on the direction of moisture change) the age of respired $CO_2$ in soils subjected to controlled manipulations."

- *Line 93: replace 'their' by 'the'*

We changed as suggested.

- *Line 94: The part about the model seems to have been thrown in at the end without much explanation. Why does the model help the interpretation? What are the challenges of understanding age and transit time and how does the model tackle that? The model results are a big part of the results section.*

We elaborated on the relevance of the model for the interpretation of our results along the manuscript to explain the connection between the experimental and the modelling approaches as well as to explain its advantages as follows:

"In addition, we used a mathematical model to better interpret the interaction between decomposition rates change, expressed through internal dynamics of the soil ($k, \alpha, \gamma$ and number of pools) and the $\Delta^{14}C$ values by targeting the range found in the incubations. Thus, our observations and models strengthen each other to gain a deeper comprehension on the relationship between soil carbon stability, $\Delta^{14}C$, age and transit time."

*Methods*

- *Please provide coordinates for sites.*

We added the missing coordinates for the Nam Co site.

- *How was the soil collected, with an auger, or another method?*

We specified the method for both of the sites.

- *The site and soil parameters would be nicer if presented on a table to so both sites can be easily compared.*

As the same information could not be recovered for both sites: CEC, pH and EC are missing for the Zoige peatland, we think that a table would burden the readability of the method section and would not contribute significantly to a proper comparison.

- *Line 129: These are analytical replicates, not field replicates?*

That is correct, these are analytical replicates, meaning that the same treatment was repeated in 3 – 6 incubation bottles to avoid ending up with few radiocarbon data due to eventual failures in gas extraction, $CO_2$ separation, C graphitization, etc.

We specified in the description of the replicates that they are analytical.

- *Table 1 has an exclamation point that seems out of place*

Corrected

- *The methods are missing information on statistical analyses to discuss the differences in the respired and bulk 14CO2 in the results. This is a big missing component that seriously concerns me, especially since there are issues in the interpretation of the results.*

We inserted the next paragraph in the methods section (Section 2.3):

"The effects of soil moisture and temperature manipulation on $\Delta^{14}C$ were evaluated for the bulk and $CO_2$ fractions in each ecosystem separately through two-way ANOVA tests (type III). This type of ANOVA is also referred to as Partial Sum of Squares and is appropriate for unbalanced data since it does not depend on the sampling structure or the particular order in the model (Shaw and Mitchell-Olds, 1993); hence, this approach adjusts best to our data set where treatments did not have equal amount of values. As for the $CO_2$ fluxes, rates were measured regularly for every treatment and mean $CO_2$ respiration rates (mg $CO_2$ g soil -1 day -1) were calculated based on the total duration of the incubation (Fig. A1)."

*Results*

- *Line 217: Remove duplicate 'to' and add comma before 'which'*

Corrected

- *Line 219: Was the difference, or maybe lack thereof, statistically significant?*

We reformulated this sentence as follows:

"In contrast, for the grassland soil, the $\Delta^{14}C$ of bulk soil (21.1 to 73.9 ‰, mean=43.3, n=33) fell similarly around the 1:1 line compared to the $\Delta^{14}CO_2$ (13.9 to 83.4 ‰,

mean=38.5, n=33, including outliers of -227.1 and -105.1) indicating that the samples behaved mostly as a well-mixed homogeneous system (type I)."

- *Figure 3: Are these results for all moisture and temperature levels? Why combine them? Having a graph that shows the effect of moisture would aid in the interpretation of the results. Figure 5 is not appropriate to show the significant result of the moisture effect on the 14CO2 signature since it splits it by temperature levels, and there were no significant temperature and moisture interactions.*

  Yes, the results are the combination of all moisture and temperature treatments. This Figure 3 allows to understand the magnitude of the difference between $\Delta^{14}C$ values between the respired $CO_2$ and the bulk soil for each ecosystem, therefore its relevance. The specific effect of moisture can also be clearly seen in Figure 5 if the boxplots (60 and 95 WFPS) levels are taken for each temperature individually. From this Figure, we can conclude that the $\Delta^{14}C-CO_2$ values in peatlands are more depleted as moisture increases.

- *Line 227: Please add the p-value for the CO2 model and discuss the results as significant or not. The statistical model needs to be added to the methods.*

  See comment above on the methods (section 2.3).

- *Line 233: It is not clear from the stats results or the graph that temperature and moisture manipulations caused a response in the vertical and horizontal direction in the $\Delta$14C of bulk versus CO2 space. Ecosystem type seems to be the driver of the clustering, please explain how this conclusion was made.*

  Excellent point. We adjusted this affirmation as follows (section 3.2):

  "Changes in the vertical and horizontal direction of the $\Delta^{14}C$-bulk versus $\Delta^{14}C-CO_2$ space is more evident across ecosystem type, which at the same time implies specific environmental conditions for the stability of SOM."

*Discussion*

- *Line 291: Please discuss the moisture results as whether they are significant or not for each ecosystem. Additionally, since there was not a significant moisture x temperature interaction, it is not accurate to discuss the moisture effect on different temperature treatments. It is also unclear whether the sites were added in the model structure; therefore, I'm not sure if the moisture effect can even be discussed separately for each site. All of this needs to be addressed in the methods and results sections.*

  We modified this section to discuss treatment variation in terms of significance as follows:

  "Nevertheless, changes in WFPS had a significant effect on the $\Delta^{14}CO_2$ and $\Delta^{14}C$-bulk of grassland soils and only on the $\Delta^{14}CO_2$ of peatland soils."

  Besides, the contrasting conditions between grassland and peatland were adapted in the model by differentiating the ranges of decomposition rates in both slow (peatland) and fast (grassland) cycling systems. This was added in the section 2.4 of the methods and 3.2 of the results.

Since SOM stability depends on the direction of moisture change, we consider that approaching to this discussion has to differentiate grasslands from peatlands as SOM would react different when moisture thresholds are crossed. Therefore, we emphasize on the importance of understanding not only the values of WFPS but also the direction in which it changes (e.g., from dry to wet or wet to dry) as well as its interplay with temperature as modulator of SOM decomposition rates.

New references:

Pegoraro, E. F., Mauritz, M. E., Ogle, K., Ebert, C. H., & Schuur, E. A. (2021). Lower soil moisture and deep soil temperatures in thermokarst features increase old soil carbon loss after 10 years of experimental permafrost warming. Global change biology, 27(6), 1293-1308.

Shaw, R. G., & Mitchell-Olds, T. (1993). ANOVA for unbalanced data: an overview. Ecology, 74(6), 1638-1645.

---

## Referee Report (RR1)

**Reviewer comment to Authors**

Angarife-Escobar et al studied the effect of temperature and moisture on $CO_2$ respiration to understand the stability of carbon in peatland and grassland. The paper presents interesting insights and is clearly well written. Nonetheless, I have few observations.

**General comments**

Throughout the paper the Authors refer to heterotrophic $CO_2$ diffusion rate from the soil as $CO_2$ respiration. Please note that you did not measure $CO_2$ respiration but measured the diffusion rates of $CO_2$ from the soil. I am aware that many Scientists use this terminology but is not entirely correct as not all the respired $CO_2$ diffuses out of the soil. I would advise to acknowledge this in the beginning that you measured $CO_2$ diffusion rate as a close proxy of $CO_2$ respiration.

Further note that you did not measure autotrophic or total $CO_2$ diffusion rate but the incubation performed measured heterotrophic $CO_2$ diffusion rate. It is useful to to refer to your respiration rate as heterotrophic $CO_2$ respiration.

**Specifics**

**Line 141: Incubations for each subset ended simultaneously until every sample had an estimated concentration of $CO_2$-C in the headspace equivalent to ≥2mg of C, enough for radiocarbon analysis.**

This statement is not clear to me. Further for how long did it take to achieve a $CO_2$ flux of ≥2 mg of C ? Was there pre-incubation period?

**144. For sampling headspace air, 50-ml vials were filled with 12 g of soil (± 1.5 g) and placed inside 0.5 L glass flasks along with 0.2 ml of water at the bottom of the flask (away from contact with the sample) to avoid possible drying (Dioumaeva et al., 2002); thereafter the flasks were sealed with rubber plugs and screwed with plastic caps. Flasks with samples were flushed with synthetic air ($CO_2$ free) to remove atmospheric $CO_2$. This flushing marked the starting day of the incubations.**

How did you make sure the disturbed soils were repacked to a bulk density similar to that of undisturbed field soils?

The headspace volume is not mentioned. What headspace volume were left when the 12g of soil were packed in 50 mL. Were the headspace left uniform for all samples throughout? Were headspace volume corrected for in the flux calculations?

The $CO_2$ respiration was measured within what time interval? every minute, 10 minutes or what exactly?

How were the $CO_2$ concentrations converted into fluxes? This should be stated in the methods.

**Results**

**Fig. 4**: The symbols for WFPS at 60 and 95 % are not visible in the graph.

At such a high WFPS of 95%, doesn't C emission shift to $CH_4$ pathway rather than $CO_2$?

**Fig 5 C and D**: You measured the $CO_2$ respiration at four different temperatures. With this result you can derive and compare an important parameter of C transformation. i.e. the coefficient of temperature sensitivity of $CO_2$ respiration $Q_{10}$.

---

## Author Response (AR2)

**RC2**: 'Comment on egusphere-2023-210', Additional anonymous Referee #3, 9 January 2024

We would like to extend our sincere appreciation to Referee #3 for diligently reviewing our manuscript. We have thoroughly addressed all the comments and have integrated our responses to the previous referees' feedback. The key points from Referee #3 review are summarized below in italics, with our reply in normal font.

**Reviewer comment to Authors**

• Angarife-Escobar et al studied the effect of temperature and moisture on CO2 respiration to understand the stability of carbon in peatland and grassland. The paper presents interesting insights and is clearly well written. Nonetheless, I have few observations.

**General comments**

Throughout the paper the Authors refer to heterotrophic CO2 diffusion rate from the soil as CO2 respiration. Please note that you did not measure CO2 respiration but measured the diffusion rates of CO2 from the soil. I am aware that many Scientists use this terminology but is not entirely correct as not all the respired CO2 diffuses out of the soil. I would advise to acknowledge this in the beginning that you measured CO2 diffusion rate as a close proxy of CO2 respiration.

We modified and added as recommended:

The rate of accumulation of  $CO_2$  in the headspace of our incubations represents the diffusion rate of heterotrophic  $CO_2$  respiration released from the incubated soils. We refer throughout the manuscript to heterotrophic respiration, but we acknowledge that our measurements better capture how this heterotrophic  $CO_2$  respiration flux diffuses out of the soil.

• Further note that you did not measure autotrophic or total CO2 diffusion rate but the incubation performed measured heterotrophic CO2 diffusion rate. It is useful to to refer to your respiration rate as heterotrophic CO2 respiration.

We changed the terminology to heterotrophic  $CO_2$  respiration instead of soil respiration along the manuscript.

**Specifics**

 Line 141: Incubations for each subset ended simultaneously until every sample had an estimated concentration of CO2 -C in the headspace equivalent to ≥2mg of C, enough for radiocarbon analysis. This statement is not clear to me. Further for how long did it take to achieve a CO2 flux of ≥2 mg of C ? Was there pre-incubation period?

We modified the sentence to avoid ambiguity:

Incubations for each subset concluded concurrently once all the samples reached a C concentration (from  $CO_2$ ) in the headspace estimated to be equal to or exceeding 2 mg, sufficient for subsequent radiocarbon analysis. This approach was not possible in

two of the 16 subsets due to lab material limitations and therefore, grassland samples were incubated between 15 and 67 days, while peatland samples were incubated for 13 days (Table A1).

Additionally, we added for clarification:

We conducted two sets of incubation experiments without pre-incubation period, one set with the grassland soil and a second set with the peatland soil.

144. For sampling headspace air, 50-ml vials were filled with 12 g of soil (± 1.5 g) and placed inside 0.5 L glass flasks along with 0.2 ml of water at the bottom of the flask (away from contact with the sample) to avoid possible drying (Dioumaeva et al., 2002); thereafter the flasks were sealed with rubber plugs and screwed with plastic caps. Flasks with samples were flushed with synthetic air (CO2 free) to remove atmospheric CO2. This flushing marked the starting day of the incubations. How did you make sure the disturbed soils were repacked to a bulk density similar to that of undisturbed field soils?

Unfortunately, since the soils were disturbed, we did not try to match an exact value of bulk density during the repacking.

• The headspace volume is not mentioned. What headspace volume were left when the 12g of soil were packed in 50 mL. Were the headspace left uniform for all samples throughout? Were headspace volume corrected for in the flux calculations?

We appreciate this helpful comment. We have now corrected the flux calculation considering the change of headspace when adding the soil. Consequently, we added the following in the methodology:

The headspace volume of the incubation flasks was measured as 587 ml, which was corrected after adding the soil. In average, the final headspace was 575.6 and 533 ml for the incubated grassland and peatland soils, respectively. These values were used to calculate the fluxes of heterotrophic  $CO_2$  respiration.

Additionally, we modified Figure 5:

 The CO2 respiration was measured within what time interval? every minute, 10 minutes or what exactly?

Although we measured fluxes at several time intervals for the different subsets of incubation, the fluxes calculation was made based on the last measurement of heterotrophic  $CO_2$  concentration. Here we complemented this information by adding the column "flux duration (day)" in Table A1, which is referenced along the manuscript when referring to the flux accumulation.

| Fable | A1. | Mean | daily | $CO_2$ | respiration | for ind | cubated | soils | under | temperat    | ture and | I WFPS | variation. |
|-------|-----|------|-------|--------|-------------|---------|---------|-------|-------|-------------|----------|--------|------------|
|       |     |      |       |        |             |         |         |       |       | · · · · · · |          |        |            |

| Ecosystem   | Temperature
(°C) | WFPS
(%) | Mean CO 2 respiration
(mg CO 2 g soil $^{-1}$ day $^{-1}$ ) | σ     | Flux duration
(day) | Incubation time (day)
n = number of samples |
|-------------|---------------------|-------------|--------------------------------------------------------------------------------------|-------|------------------------|------------------------------------------------|
|             | 20                  | 95          | 0.044                                                                                | 0.009 | 19                     | 2n = 19, 1n = 32                               |
|             | 20                  | 60          | 0.028                                                                                | 0.003 | 19                     | 4n = 19                                        |
|             | 15                  | 95          | 0.023                                                                                | 0.004 | 15                     | 3n = 30, 1n = 15                               |
| Constant    | 15                  | 60          | 0.016                                                                                | 0.003 | 15                     | 4n = 30                                        |
| Grassland   | 10                  | 95          | 0.011                                                                                | 0.001 | 60                     | 3n = 66                                        |
|             | 10                  | 60          | 0.007                                                                                | 0.001 | 60                     | 6n = 66                                        |
|             | 5                   | 95          | 0.005                                                                                | 0.000 | 60                     | 5n = 67                                        |
|             | 5                   | 60          | 0.004                                                                                | 0.001 | 60                     | 4n = 67                                        |
|             | 20                  | 95          | 0.462                                                                                | 0.032 | 7                      | 5n = 13                                        |
|             | 20                  | 60          | 0.337                                                                                | 0.014 | 7                      | 4n = 13                                        |
|             | 15                  | 95          | 0.299                                                                                | 0.007 | 7                      | 5n = 13                                        |
| De etlere d | 15                  | 60          | 0.212                                                                                | 0.004 | 7                      | 3n = 13                                        |
| Peatland    | 10                  | 95          | 0.189                                                                                | 0.010 | 9                      | 5n = 13                                        |
|             | 10                  | 60          | 0.121                                                                                | 0.020 | 9                      | 5n = 13                                        |
|             | 5                   | 95          | 0.061                                                                                | 0.004 | 7                      | 4n = 13                                        |
|             | 5                   | 60          | 0.040                                                                                | 0.001 | 7                      | 5n = 13                                        |

We included the explanation on the role of time interval of the heterotrophic  $\text{CO}_2$  fluxes in the comment below.

• How were the CO2 concentrations converted into fluxes? This should be stated in the methods.

We added in the methodology:

Rates were measured at intervals of 1 to 2 weeks using a  $CO_2$  analyzer LI-COR 6262 for every treatment and mean heterotrophic  $CO_2$  respiration rates (mg  $CO_2$  g soil-1 day-1) were calculated through the division of  $CO_2$  concentration in the headspace by the product of the accumulation duration (days) (Table A1) and the mass of the introduced soil (g).

**Results**

• Fig. 4: The symbols for WFPS at 60 and 95 % are not visible in the graph.

The symbols for the samples in Fig 4 indicate a combined treatment, where WFPS is represented by shape and temperature by color. Thus, the shapes for WFPS can be seen in the graph in different colors depending on the temperature. To deal with the ambiguity of the black color in WFPS, we have modified the Fig 4 and changed the legend to empty shapes.

• At such a high WFPS of 95%, doesn't C emission shift to CH4 pathway rather than CO2?

As we had mentioned in the reply to referee#1 and 2, "95% of WFPS is certainly a high level of moisture saturation, nonetheless, the incubated soil had still wide contact with oxygen, allowing SOM oxidation and  $CO_2$  accumulation." Unfortunately, we did not measure  $CH_4$  levels and cannot elaborate on this topic.

We added in the methodology:

Despite the high WFPS, the soil samples had still contact with air inside the vials, which guaranteed microbial decomposition of the organic matter and accumulation of heterotrophic CO2 respiration.

• Fig 5 C and D: You measured the CO2 respiration at four different temperatures. With this result you can derive and compare an important parameter of C transformation. i.e. the coefficient of temperature sensitivity of CO2 respiration Q10.

Although the  $Q_{10}$  parameter has been widely used as an indicator of biochemical processes, we consider it is not useful for our approach due to several reasons. First, there is a large debate on the usefulness of  $Q_{10}$  as a metric of temperature sensitivity of SOM. Although  $Q_{10}$  is conceptually clearly defined, there are different formulas to calculate it (Fang et al., 2005, Fierer et al., 2005, Conant et al., 2008, Wetterstedt et al., 2010) that impose limitations and even biases on their intercomparison.

Furthermore, various methods frequently employed to assess the temperature sensitivity of different substrates can produce contradictory results, despite being based on the same fundamental principles (Sierra 2012). For example, theoretical analysis on the "quality-temperature" hypothesis (Bosatta & Ågren 1999) have been contradicted by empirical mixed results with no conclusive evidence (von Lützow and Kögel-Knabner 2009). In consequence, some authors have discouraged the use of  $Q_{10}$ s (Davidson *et al.*, 2006, Sierra 2012) and we prefer not to perpetuate the use of this metric. Second, our methodological setting intended to measure C concentrations from heterotrophic respiration for 14C analysis instead of cond